# Nde1 promotes Lis1-mediated activation of dynein

Yuanchang Zhao [1,2], Sena Oten[2] & Ahmet Yildiz [1,2,3] ✉

Cytoplasmic dynein drives the motility and force generation functions towards the microtubule minus end. The assembly of dynein with dynactin and a cargo adaptor in an active transport complex is facilitated by Lis1 and Nde1/Ndel1. Recent studies proposed that Lis1 relieves dynein from its autoinhibited conformation, but the physiological function of Nde1/Ndel1 remains elusive. Here, we investigate how human Nde1 and Lis1 regulate the assembly and subsequent motility of mammalian dynein using in vitro reconstitution and single molecule imaging. We find that Nde1 recruits Lis1 to autoinhibited dynein and promotes Lis1-mediated assembly of dynein-dynactin adaptor complexes. Nde1 can compete with the α2 subunit of platelet activator factor acetylhydrolase 1B (PAF-AH1B) for the binding of Lis1, which suggests that Nde1 may disrupt PAF-AH1B recruitment of Lis1 as a noncatalytic subunit, thus promoting Lis1 binding to dynein. Before the initiation of motility, the association of dynactin with dynein triggers the dissociation of Nde1 from dynein by competing against Nde1 binding to the dynein intermediate chain. Our results provide a mechanistic explanation for how Nde1 and Lis1 synergistically activate the dynein transport machinery.

Cytoplasmic dynein-1 (dynein hereafter) drives retrograde transport of a wide variety of intracellular cargos, including membranous organelles, vesicles, mRNA, and unfolded proteins. Dynein also plays essential roles in cell division, including nuclear envelope breakdown, focusing the mitotic spindle, and transporting spindle assembly checkpoint signals[1]. Mutations in dynein and its regulatory proteins have been linked to severe developmental and neurological disorders, including spinal muscular atrophy, motor neuron degeneration, ALS, and schizophrenia[2].

The dynein complex is composed of a homodimer of dynein heavy chains (DHC) and several intermediate, light-intermediate, and light chains associated with DHC. Motility is driven by the C-terminal motor domain of DHC, which contains a catalytic ring of six AAA subunits, a microtubule-binding domain, and other mechanical elements that drive minus-end directed motility[3]. Similar to kinesin[4], isolated dynein remains in an autoinhibited conformation through direct interactions between its two motor domains[5,6]. Dynein motility is activated by its association with a multi-subunit complex, dynactin,

and the coiled-coil domain of an activating adaptor that links the motor to its cellular cargo (DDA complex)[7–10].

The assembly, activation, and subsequent motility of the dynein transport machinery are highly regulated by accessory proteins, Lis1 and Nde1/Ndel1[11]. Lis1 is the only known protein that directly binds to the motor domain of dynein and is required for virtually all dynein functions in the cytoplasm of eukaryotic cells[11,12]. Mutations in the *LIS1* gene have been shown to disrupt many dynein-driven processes in cells and heterozygous mutations to *LIS1* cause the brain neurodevelopmental disease, lissencephaly[11]. Lis1 forms a homodimer through its N-terminal LisH domain[13,14] and binds to the AAA+ ring and stalk of dynein through its β-propeller domains[15]. Recent in vitro studies proposed that Lis1 binding to the dynein motor domain is incompatible with self-interactions between the two motor domains in the autoinhibited phi conformation, thereby rescuing dynein from autoinhibition and promoting the formation of the DDA complex[16–19]. After the initiation of transport, Lis1 is not required for processive motility and its dissociation from DDA has been reported to result in faster

[1]Physics Department, University of California, Berkeley, CA 94709, USA. [2]Department of Molecular and Cell Biology, University of California, Berkeley, CA 94709, USA. [3]Biophysics Graduate Group, University of California, Berkeley, CA 94709, USA. ✉e-mail: yildiz@berkeley.edu

motility[16–18,20]. This model are compatible with studies of Lis1 in live cells[21–28] and provides a mechanistic explanation for why Lis is required for dynein-mediated transport.

Nde1 and Ndel1 are highly conserved proteins that play important roles in the dynein-driven transport of intracellular cargos and nuclear oscillations in developing neurons, as well as dynein-mediated functions in mitosis[29]. Nde1/Ndel1 contains an N-terminal coiled-coil domain that interacts with the dynein intermediate chain (DIC) and the β-propeller domain of Lis1, whereas the C-terminus is mostly disordered[30–33]. While Lis1 deletion is deleterious for most dynein-driven processes[23,34], Nde1 or Ndel1 deletion results in relatively milder phenotypes[23,35,36], possibly due to overlapping functions of Nde1 and Ndel1. However, co-depletion of Nde1 and Ndel1, or Nde1/Ndel1 and Lis1 have been shown to severely impair retrograde transport[23,35]. While the phenotype caused by Nde1 deletion could be rescued by overexpression or exogenous addition of Lis1, Lis1 deletion could not be rescued by Nde1 addition, demonstrating that Nde1 function is dependent on Lis1[37,38].

The mechanism by which Nde1/Ndel1 (Nde1 hereafter) regulates the dynein activation pathway together with Lis1 is not well understood. Studies in live cells suggested that Nde1's primary function is to tether Lis1 to dynein, increasing its apparent affinity for dynein[38–40]. Consistent with this model, the Nde1 mutant that cannot bind dynein failed to rescue a mitotic phenotype caused by Nde1 depletion[41]. Similarly, the expression of a dynein mutant that cannot adopt the phi conformation partially rescued defects in Nde1 depletion in filamentous fungi[19]. Nde1 overexpression rescues phenotypes caused by depletion, but not deletion, of Lis1[35,42]. However, strong overexpression or the addition of excess Nde1 has been shown to cause a dominant negative effect on dynein-driven processes in many organisms[22,38,43,44], but how excess Nde1 disrupts dynein function remained unknown.

The tethering model has been challenged by several observations made in vivo and in vitro. Overexpression of Nde1 mutant that cannot bind Lis1 was reported to rescue mitotic phenotypes of Nde1 depletion in several cell types[41,45]. In vitro studies showed that while Lis1 promotes the assembly of the DDA complex[16–18], Nde1 competes against the p150[Glued] subunit of dynactin to interact with DIC[32,46,47]. Furthermore, Lis1 increases whereas Nde1 decreases the microtubule-binding affinity of isolated dynein[12,31,48,49]. These observations indicate that Nde1's role in the dynein activation pathway is more complex than tethering Lis1 to dynein and may also involve Nde1-mediated regulation of dynein independent of Lis1. Consistent with the possibility that Nde1 and Lis1 have related but distinct roles in the dynein pathway, *NDE1* and *LIS1* mutations are linked to distinct neurodevelopmental diseases[40,50].

Early in vitro studies have reported conflicting information on whether Nde1 enhances dynein activity or inhibits it[12,31,48]. These studies were performed before it was understood that dynein remains inactive in the absence of dynactin and an adaptor protein[5,8,9,51]. To understand the role of Nde1 in the dynein activation pathway, we directly monitored the assembly and motility of the mammalian dynein-dynactin-BicDR1 (DDR) complex in the presence of human Lis1 and Nde1 in vitro. We observed that Nde1 promotes the processive motility of mammalian dynein-dynactin in the presence of Lis1. This promotive effect is through tethering of Lis1 to dynein, as Nde1 mutants that cannot bind Lis1 or dynein failed to stimulate dynein motility. In comparison, excess Nde1 inhibited dynein motility by competing against the DIC interaction site of dynactin during the assembly of the DDR complex. Nde1 was released from dynein before the initiation of DDR motility. These results illuminate the physiological function of this key regulatory protein in the dynein pathway.

## Results
### Nde1 promotes the assembly and activation of dynein together with Lis1
To investigate how Nde1 regulates dynein, we reconstituted the assembly of wild-type human dynein (wtDyn) in the presence of pig brain dynactin, and LD655-labeled mouse BicDR1 adaptor in vitro and monitored the motility of processive DDR complexes on surface-immobilized microtubules using a total internal reflection fluorescence (TIRF) imaging assay (Fig. 1a). In the absence of Lis1, we observed processive runs of the DDR complexes assembled with wtDyn (wtDDR) at low frequency, consistent with autoinhibition of dynein[6]. The addition of 10 nM unlabeled Nde1 without Lis1 did not substantially affect the frequency of processive runs, but the addition of excess (1000 nM) Nde1 almost fully inhibited wtDDR motility (Fig. 1b, c).

We next tested how Nde1 addition affects dynein motility in the presence of Lis1. Consistent with previous reports that Lis1 facilitates the assembly of the DDR complexes and increases the likelihood of dynactin to recruit two dyneins[16,17], Lis1 addition increased the run frequency of wtDDR by up to 2.6 fold and resulted in faster motility in the absence of Nde1 (Fig. 1d, e and Supplementary Fig. 1). The addition of 1–10 nM Nde1 resulted in up to 16-fold increase in run frequency in the presence of Lis1 (Fig. 1d, e, Supplementary Fig. 1a, and Supplementary Movie 1), demonstrating that Lis1 and Nde1 synergistically promote the activation of dynein motility. Nde1 addition also led to a modest increase in the average velocity of processive runs (Supplementary Fig. 1b), indicating that Nde1 may promote the recruitment of two dyneins to dynactin.

We tested whether higher concentrations of Nde1 further promote dynein motility in the presence of Lis1. Unlike this expectation, 250 nM Nde1 substantially lowered the run frequency, and 1000 nM Nde1 almost fully inhibited dynein motility in the presence of 10–250 nM Lis1 (Fig. 1d, e, and Supplementary Fig. 1a). These results show that Nde1 regulates activation of dynein in a biphasic manner in vitro: low concentration (1–10 nM) of Nde1 is sufficient to enhance dynein motility together with Lis1, whereas excess Nde1 inhibits dynein independent of Lis1, consistent with Nde1 overexpression to disrupt dynein-dependent functions in cells[22,38,43,44].

To distinguish between these two opposing effects of Nde1, we repeated our measurements using a dynein mutant unable to attain the phi conformation (mtDyn)[6]. Unlike wtDDR, DDR complexes assembled with mtDyn (mtDDR) exhibited robust motility in the absence of Lis1 and Nde1, the addition of 50 nM Lis1 and 10 nM Nde1 only slightly (1.2 fold) increased the run frequency (mtDDR, Fig. 1f, g, Supplementary Fig. 2, and Supplementary Movie 2)[16]. However, similar to wtDDR, the addition of excess Nde1 almost fully abolished mtDDR motility in the presence of Lis1 (Fig. 1f, g). These results indicate that a low concentration of Nde1 is sufficient for a more effective release of dynein from its phi conformation by Lis1, whereas inhibition of dynein by excess Nde1 is not related to the autoinhibitory mechanism of dynein. We also noticed the run frequency of wtDDR becomes nearly equivalent to that of mtDDR only in the presence of Lis1 and low concentration (10 nM) of Nde1 (Fig. 1g), suggesting that both Lis1 and Nde1 are required for efficiently rescuing wtDyn from its phi conformation.

### Nde1 tethers Lis1 to dynein
We next turned our attention to understanding how Nde1 facilitates Lis1-mediated activation of dynein motility. The N-terminal coiled-coil of Nde1 has high sequence conservation among different species and contains distinct binding sites for Lis1 and DIC[52]. The C-terminus of Nde1 is predicted to contain a short coiled-coil domain flanked with intrinsically disordered regions[53,54]. Using AlphaFold2[53,54], we modeled Lis1 binding to Nde1. The model predicted that Lis1 binds to the N-terminal coiled-coil domains of Nde1 with both of its β-propeller domains. The C-terminal coiled-coil of Nde1 folds back to the N-terminal coiled-coil domain at a position near the Lis1 binding site (Fig. 2a and Supplementary Movie 3), raising the possibility of interference of this region with Lis1 binding. Mass photometry assays confirmed that both Lis1 and Nde1 form a homodimer, and Lis1 forms a complex with Nde1 with either 1:1 or 2:1 stoichiometry (Fig. 2b).

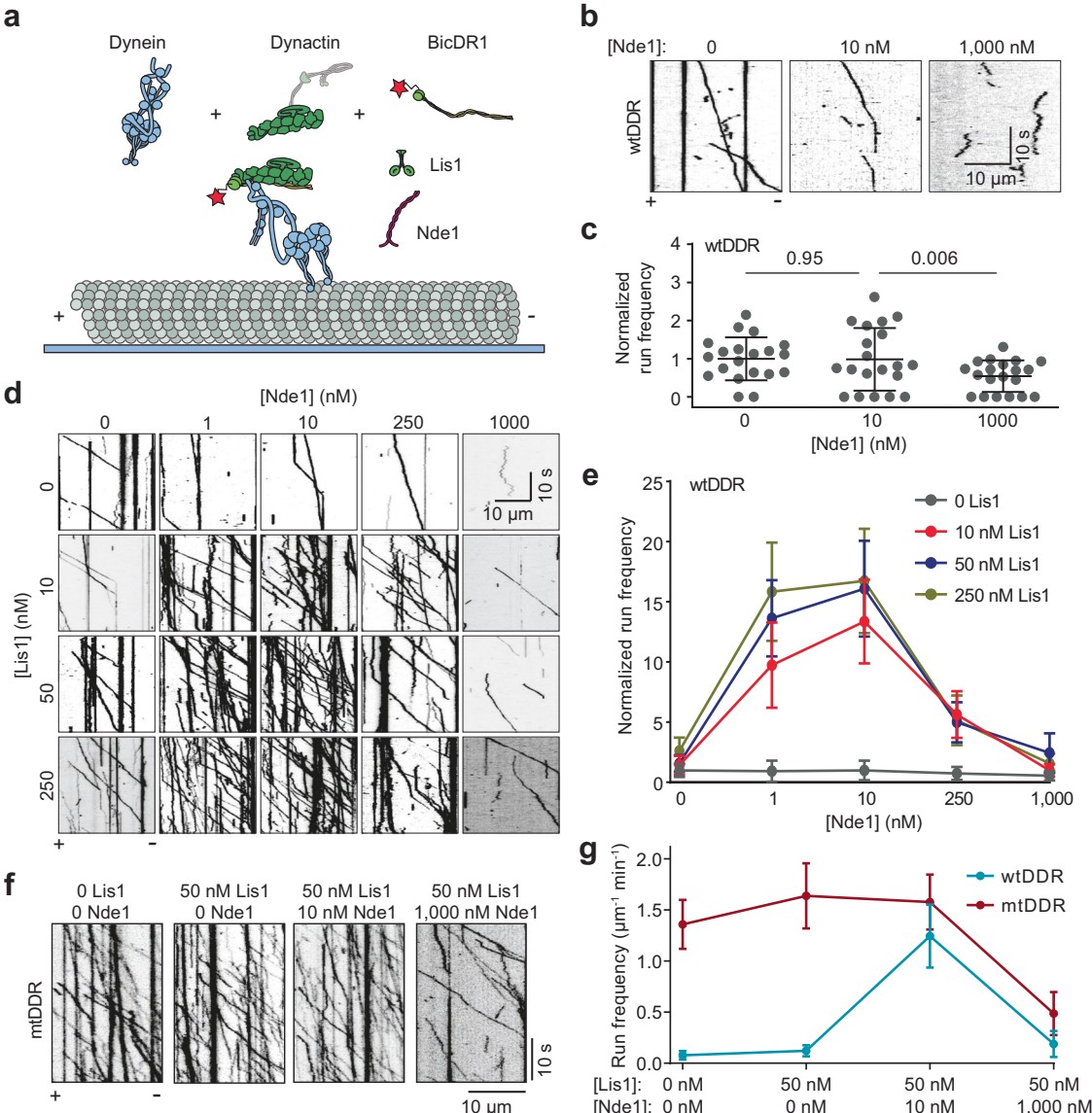

**Fig. 1 | Nde1 promotes dynein motility together with Lis1. a** In vitro reconstitution of dynein motility using dynein, dynactin, BicDR1, Lis1, and Nde1 on biotinylated microtubules immobilized to the glass surface. The red star represents an LD655 dye attached to BicDR1 for TIRF imaging. Dynein, dynactin, and tubulin were not labeled with a fluorescent dye. **b** Representative kymographs show the motility of wtDDR in the presence of 0, 10, and 1000 nM Nde1 and the absence of Lis1. **c** Normalized run frequency distribution of wtDDR under different Nde1 concentrations. The center line and whiskers represent the mean and s.d., respectively ($n = 20$ microtubules for each condition). $P$ values are calculated from a two-tailed $t$

test. **d** Representative kymographs show the motility of wtDDR in the presence of 0–250 nM Lis1 and 0–1000 nM Nde1. **e** The run frequency of wtDDR in different Lis1 and Nde1 concentrations (mean ± s.d.; $n = 20$ microtubules for each condition). Results were normalized to the 0 nM Lis1 and 0 nM Nde1 condition. **f** Representative kymographs show the motility of mtDDR complexes with or without Nde1 and Lis1. **g** Run frequencies of wtDDR and mtDDR with different Nde1 and Lis1 concentrations (mean ± s.d.; $n = 10$ microtubules for each condition). Source data are provided as a Source Data file.

Previous studies proposed that the primary role of Nde1 is to tether Lis1 to dynein[12,27,31,38,39,55]. To test this model, we immobilized Alexa488-labeled wtDyn from its tail to a coverslip and determined the colocalization of LD555-labeled Lis1 to dynein in the presence and absence of unlabeled Nde1 (Fig. 2c and Supplementary Fig. 3a). Consistent with the tethering mechanism, we observed that increasing the Nde1 concentration from 0 to 5 nM substantially increased the colocalization of Lis1 to surface-immobilized dynein (Fig. 2d). Similarly, co-immunoprecipitation (Co-IP) assays showed higher Lis1 binding to dynein under increasing concentrations of Nde1 (Supplementary Fig. 3b). We also observed that Lis1 increases the colocalization of Nde1 with surface immobilized dynein (Fig. 2d). The analysis of single-molecule colocalization events showed that both Lis1-binding and Nde1-binding to dynein are dynamic with

average bound times of 12 ± 1 and 21 ± 2 s, respectively (Fig. 2e). The addition of Nde1 increases the average bound time of Lis1 to dynein (35 ± 2 s) and Lis1 increases the average bound time of Nde1 to dynein (42 ± 2 s). Collectively, these results show that Nde1 promotes the binding of Lis1 to dynein, and Nde1 and Lis1 stabilize each other's binding to the dynein complex.

To distinguish whether the N-terminal coiled-coil of Nde1 is sufficient or the C-terminus also contributes to the regulatory role of Nde1 in the dynein activation pathway, we truncated the C-terminus of Nde1 (Supplementary Fig. 4) and determined how the N-terminus of Nde1 (Nde1[1–190]) regulates dynein motility. Mass photometry assays showed that Nde1[1–190] forms a homodimer[30,56] and interacts with a Lis1 dimer at 1:1 or 1:2 ratios (Fig. 2f). Nde1[1–190] interacted with Lis1 more efficiently than Nde1 (55% versus 23% of the total population), indicating that the

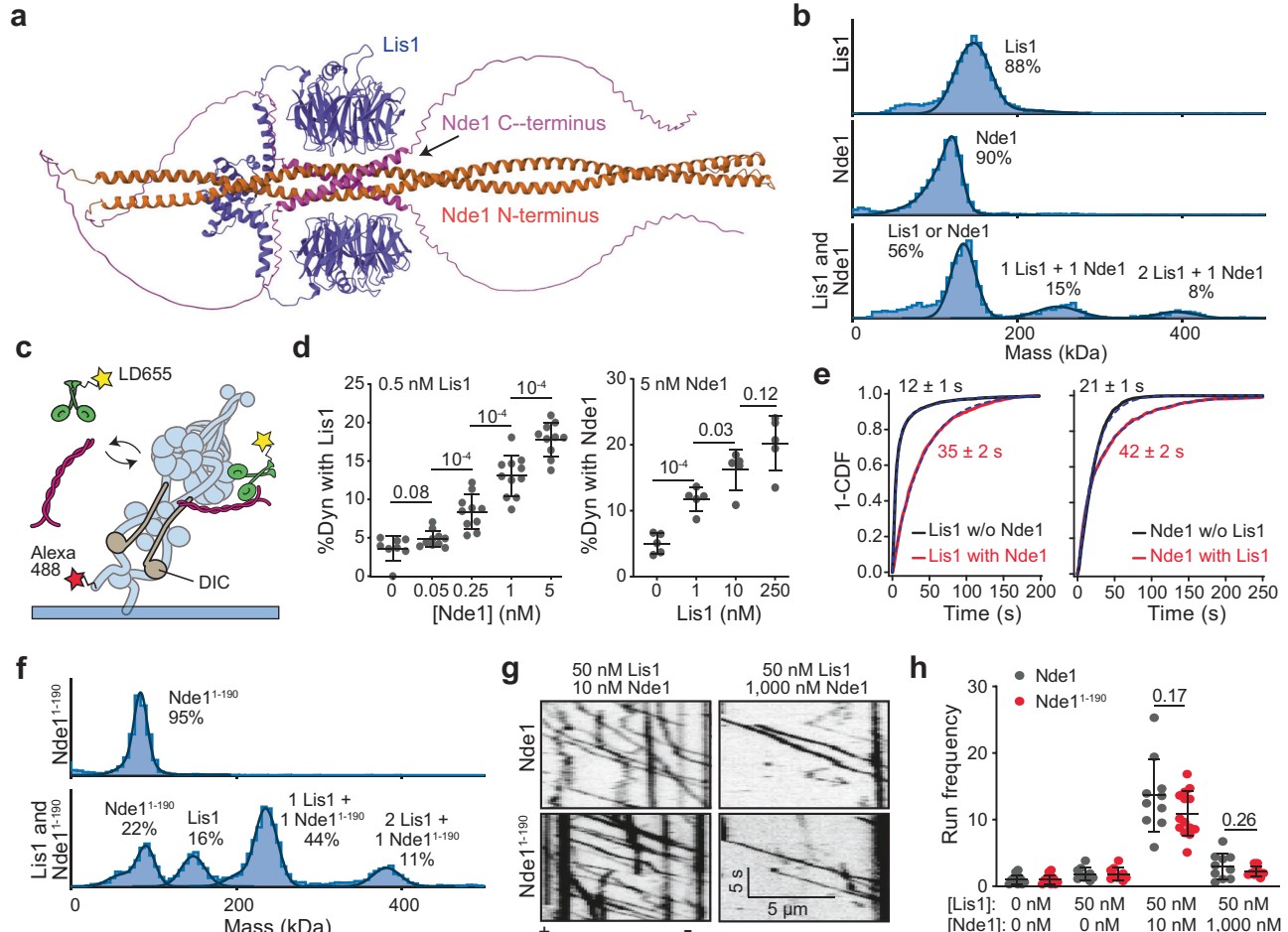

**Fig. 2 | The N-terminal coiled-coil of Nde1 tethers Lis1 to dynein and stimulates dynein motility. a** AlphaFold2 prediction of Lis1 (purple) binding to Nde1 (orange: N-terminal, pink: C-terminal). **b** Mass photometry profiles for Lis1 binding to Nde1. **c** Schematics of single molecule colocalization assay. wtDyn was labeled with both biotin and Alexa488 and immobilized on a glass surface. 0.5 nM LD555-labeled Lis1 and different concentrations of unlabeled Nde1 were flown into the chamber for observation of Lis1 colocalization to surface-immobilized dynein. **d** Percent colocalization of surface-immobilized dyneins with LD555-Lis1 under different concentrations of Nde1 (Left) and with LD655-Nde1 under different concentrations of Lis1 (Right). From left to right, $n = 8, 10, 10, 11, 10, 5, 5, 5$, and 5 imaging areas ($40\,\mu m \times 40\,\mu m$) with at least 100 immobilized dynein spots. **e** (Left) The inverse cumulative distribution function (1 - CDF) of Lis1 binding to dynein in the presence

and absence of Nde1. (Right) 1-CDF of Nde1 binding to dynein in the presence or absence of Lis1. Dashed curves represent a fit to an exponential decay to estimate the average residence time ($\pm$s.e.). **f** Mass photometry profiles for Lis1 binding to Nde1$^{1-190}$. **g** Representative kymographs show wtDDR motility in the presence of Nde1 or Nde1$^{1-190}$. **h** The run frequency distribution of wtDDR under different Lis1 and Nde1 or Nde1$^{1-190}$ concentrations. Results were normalized to the 0 nM Lis1 and 0 nM Nde1 condition. From left to right, $n = 14, 14, 10, 10, 10, 14, 10$, and 10 microtubules. In (**b**, **f**), solid curves represent a fit to multiple Gaussians to predict the average mass (Supplementary Table 2) and percentage of each population. In (**d**, **h**), the center line and whiskers represent the mean and s.d., respectively. $P$ values are calculated from a two-tailed $t$ test. Source data are provided as a Source Data file.

Nde1 C-terminus negatively regulates Lis1 binding to Nde1 (Fig. 2f). Motility assays showed that Nde1$^{1-190}$ regulated wtDDR motility similar to full-length Nde1 (Fig. 2g, h), indicating that the N-terminal coiled-coil is sufficient for both the activating and inhibitory effects of Nde1 on dynein motility in vitro.

**Dynein and Nde1 have overlapping binding sites on Lis1**
The tethering mechanism proposes that Lis1, Nde1, and dynein form a ternary complex. However, our Alphafold2 model predicts that Lis1 interacts with Nde1 through the same surface of its β-propeller domain that interacts with dynein (Fig. 2a)[15,57], raising doubts on whether Lis1 can simultaneously interact with dynein and Nde1. To test this possibility, we expressed a well-established dynein-binding mutant of Lis1 (R316A and W340A mutations on the β-propeller domain, mtLis1, Fig. 3a)[17]. These mutations did not affect Lis1 dimerization but fully disrupted Lis1 binding to Nde1$^{1-190}$ in mass photometry assays (Fig. 3b). These mutations also disrupted Lis1 binding to dynein and Nde1 in

both single-molecule colocalization and Co-IP assays (Fig. 3b, c, Supplementary Fig. 5a), demonstrating that dynein and Nde1 have overlapping binding sites on Lis1. mtLis1 also failed to enhance wtDDR motility in the presence or absence of Nde1 (Supplementary Fig. 5b, c), underscoring that binding of the β-propeller domain of Lis1 to dynein is essential for Lis1-mediated activation of dynein.

A previous study reported that Nde1's interaction with the Lis1 β propeller domain is not sufficient for the stable binding of Nde1 to Lis1[14]. Our AlphaFold2 model also predicted that the helix that connects the LisH and β propeller domains of Lis1 (amino acids 58–82) forms favorable interactions with the coiled-coil of Nde1 (Fig. 2a). We tested this possibility by truncating the N-terminus of Lis1 at two positions (Lis1$^{39-410}$ and Lis1$^{83-410}$, Fig. 3a, Supplementary Fig. 5d). Mass photometry confirmed that Lis1$^{39-410}$ is only weakly (7%) dimerized and Lis1$^{83-410}$ is a monomer (Fig. 3b). Consistent with a previous observation[14], mass photometry did not detect complex formation between Nde1 and Lis1$^{83-410}$ (Fig. 3b). In addition, unlike full-length Lis1

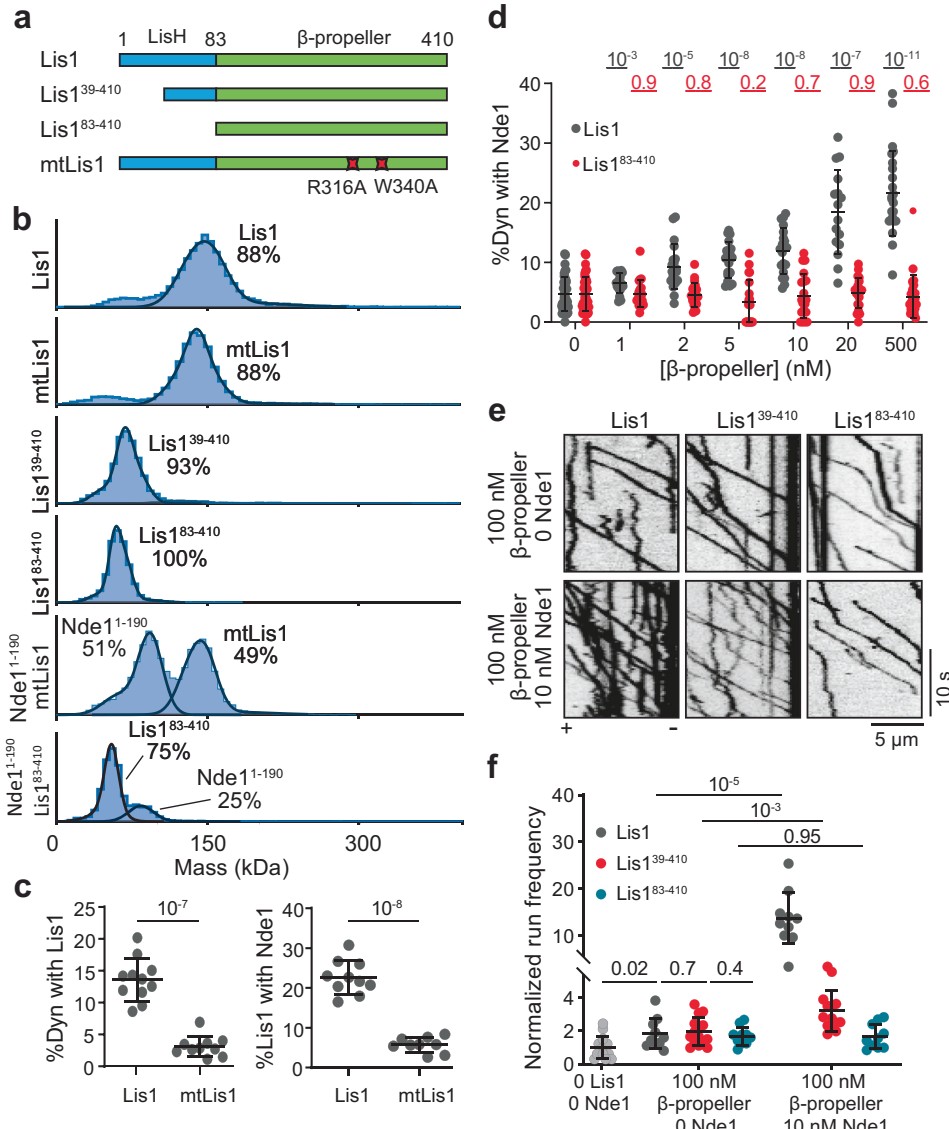

**Fig. 3 | Nde1 and dynein share the same binding site on Lis1. a** Schematics of N-terminal truncated Lis1 constructs, and a dynein binding mutant of Lis1 (mtLis1). **b** Mass photometry shows Lis1 and mtLis1 form a dimer, but Lis1[39–140] and Lis1[83–140] are monomers. Nde1[1–190] does not form a complex with mtLis1 or Lis1[83–140]. Solid curves represent a fit to multiple Gaussians to predict the average mass (Supplementary Table 2) and percentage of each population. **c** (Left) Percent colocalization of Alexa488-dynein with 5 nM LD555-Lis1 or mtLis1 and (Right) colocalization of Alexa488-labeled surface-immobilized Lis1 with 5 nM LD655-Nde1 in solution. From left to right, $n = 11, 10, 10$, and 9 imaging areas (40 μm × 40 μm) with at least 100 immobilized dynein spots. **d** Percent colocalization of Alexa488-dynein with LD655-

Nde1 under increasing concentrations of Lis1 or Lis1[83–410] ($n = 39, 39, 20, 20, 20, 20, 20, 20, 20, 20, 16, 20, 24, 20$ imaging areas (40 μm × 40 μm) with at least 100 immobilized dynein spots for each condition). Nde1 concentration was kept at 5 nM. **e** Representative kymographs show the motility of wtDDR with 100 nM Lis1 β-propellers and 0–10 nM Nde1. **f** The run frequency distribution of wtDDR with 0–100 nM Lis1 β-propellers and 0–10 nM Nde1. From left to right, $n = 14, 10, 15, 10, 10, 14$, and 10 microtubules respectively. Results were normalized to the 0 nM Lis1 and 0 nM Nde1 condition. In (**c**, **d**, **f**), the center line and whiskers represent the mean and s.d., respectively. *P* values are calculated from a two-tailed *t* test. Source data are provided as a Source Data file.

that promotes the colocalization of Nde1 with dynein, Lis1[83–410] did not promote Nde1[1–190] association with dynein (Fig. 3d). Motility assays revealed that, in the absence of Nde1, 100 nM Lis1[39–410] or Lis1[83–410] increased the run frequency of wtDDR by ~2-fold, comparable to 50 nM full-length Lis1 dimer (Fig. 3e, f), underscoring that a Lis1 β-propeller is sufficient to enhance activation of dynein motility[17]. While the addition of 10 nM Nde1 boosts the wtDDR run frequency with full-length Lis1 by more than 10-fold, Nde1 addition resulted in a modest (3.2-fold) increase in run frequency with Lis1[39–410] and had no significant effect on dynein run frequency when added together with Lis1[83–410] (Fig. 3e, f). Collectively, these results show that the stable binding of Nde1 requires its interaction with the N-terminus and the β propeller of Lis1[14].

## Excess Nde1 inhibits DDR formation by competing with dynactin for DIC

We next investigated the inhibitory effect of Nde1 on dynein motility. Previous studies have shown that the N-terminal coiled-coil of Nde1 and the p150[Glued] subunit of dynactin have overlapping DIC binding sites[32,39,40,46,47]. While the physiological significance of the p150[Glued]-DIC interaction remains elusive, it is possible that excess Nde1 competes against p150[Glued] binding to DIC and disrupts the assembly of active DDR complexes in vitro. To test this possibility, we added excess Nde1 during or after we mixed wtDyn, dynactin, and BiDR1 and tested wtDDR motility. The addition of Nde1 during DDR assembly resulted in a substantial decrease in run frequency, while Nde1 addition after the assembly did not have a significant effect on wtDDR run frequency

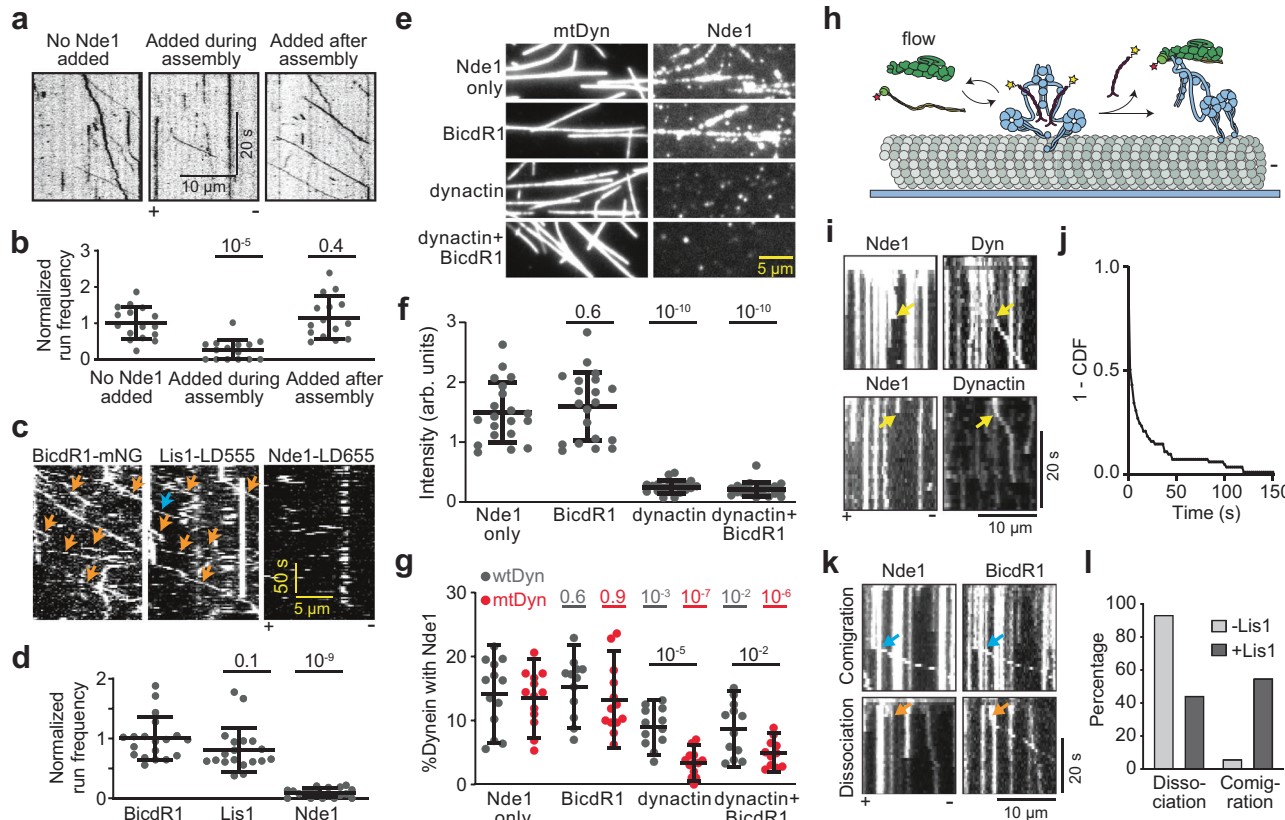

**Fig. 4 | Nde1-binding is incompatible with DDR assembly and is released before processive motility. a** Kymographs show wtDDR motility with Nde1 added during or after DDR assembly. **b** Normalized run frequency of wtDDR with Nde1 added during or after the assembly. From left to right, $n = 16$, 15, and 15 microtubules, respectively. **c** Kymographs of 100 nM BicDR1-mNG, 10 nM LD555-Lis1, and 10 nM LD655-Nde1 show comigration of Lis1 with DDR (orange arrows). Lis1 runs not colocalizing with BicDR1 (blue arrow) is due to mNG photobleaching. **d** Normalized run frequency of BicDR1-mNG, LD555-Lis1, and Nde1-LD655, respectively. $n = 19$ microtubules for each condition. **e** LD555-Nde1 landing on microtubules decorated with LD655-dynein with or without dynactin and BicDR1. **f** The normalized fluorescence intensity of Nde1 landing on dynein-decorated microtubules (mean ± s.d.; $n = 19$ microtubules for each condition). **g** Ratios of surface-immobilized wtDyn or mtDyn colocalized with 25 nM Nde1 with or without 150 nM dynactin and 50 nM BicDR1. From left to right, $n = 13$, 13, 12, 13, 12, 11, 12, 13 imaging areas (40 μm ×

40 μm) with minimum 40 dynein spots. **h** Schematics show real-time monitoring of the initiation of wtDDR motility. Microtubules were decorated with WT dynein-GFP and Nde1-LD655 in the absence of Lis1. During imaging, dynactin and BicDR1 were flown into the chamber. Either BicDR1 or dynactin was labeled with LD555. **i** Kymographs show Nde1 that colocalizes with a component of the DDR complex (dynein, BicDR1, or dynactin) disappears before the initiation of wtDDR motility (yellow arrows, three independent experiments). **j** 1-CDF of time between the disappearance of the colocalized Nde1 and the beginning of wtDDR motility. **k** Kymographs show comigration (blue arrows) or dissociation (orange arrows) of Nde1 from DDR during the initiation of dynein motility in the presence of Lis1 (Three independent experiments). **l** Percentage of Nde1 that dissociates from or comigrates with DDR with or without Lis1. In (**b**, **d**, **f**, **g**), the center line and whiskers represent the mean and s.d., respectively. *P* values are calculated from a two-tailed *t* test. Source data are provided as a Source Data file.

(Fig. 4a, b). Therefore, excess Nde1 impedes dynein motility by preventing DDR formation, rather than inhibiting the motility of preassembled complexes on microtubules.

To understand why excess Nde1 does not inhibit the complexes that are walking along microtubules, we monitored the association of fluorescently labeled Lis1 and Nde1 with dynein in single-molecule assays. If Nde1 can still associate with DIC, but its binding does not affect the motility of dynein already assembled with dynactin, we expected to observe colocalization of Nde1 to processive wtDDR complexes. While 70% of DDR complexes co-migrated with Lis1, only less than 10% comigrated with Nde1 in three-color TIRF assays (Fig. 4c, d). This observation indicates that Nde1 does not bind to dynein assembled with dynactin.

If Nde1 and dynactin binding to DIC are mutually exclusive, we anticipated dynactin binding to release Nde1 from dynein. Because dynactin has a low affinity to bind phi dynein and more readily interacts with open dynein[6], we tested this possibility for both mtDyn, which cannot form the phi conformation and remains in the open conformation[6] and wtDyn, which contains a mixture of phi (~70%) and open conformation (~30%, not shown). We first decorated surface-

immobilized microtubules with mtDyn and observed Nde1 binding to mtDyn on microtubules (Fig. 4e). The addition of dynactin, but not the cargo adaptor, caused almost all of Nde1 to be released from microtubules (Fig. 4e, f). Second, we monitored Nde1 colocalization to surface-bound dynein. The addition of dynactin reduces Nde1 colocalization to wtDyn by 37% (Fig. 4g). When we repeated the same experiment with mtDyn, dynactin addition substantially reduced the colocalization of Nde1 to dynein by 75% (Fig. 4g). These results are consistent with the idea that dynactin causes the release of Nde1 from DIC after it interacts with dynein in the open conformation.

To observe dynactin-mediated release of Nde1 from dynein in real-time, we first determined colocalization of Nde1 to dynein on microtubules, and while imaging, introduced dynactin and BicDR1 to initiate DDR motility (Fig. 4h). In the absence of Lis1, 96% of the Nde1 spots that colocalized with dynein were released upon the appearance of a dynactin signal on the same spot (Fig. 4i–l). More than half of dyneins started to move processively within 2 s after the disappearance of the colocalized Nde1 signal (Fig. 4j), indicating that dynactin binding causes Nde1 to release from dynein. When we repeated this assay in the presence of Lis1, we observed only 50% of Nde1 to

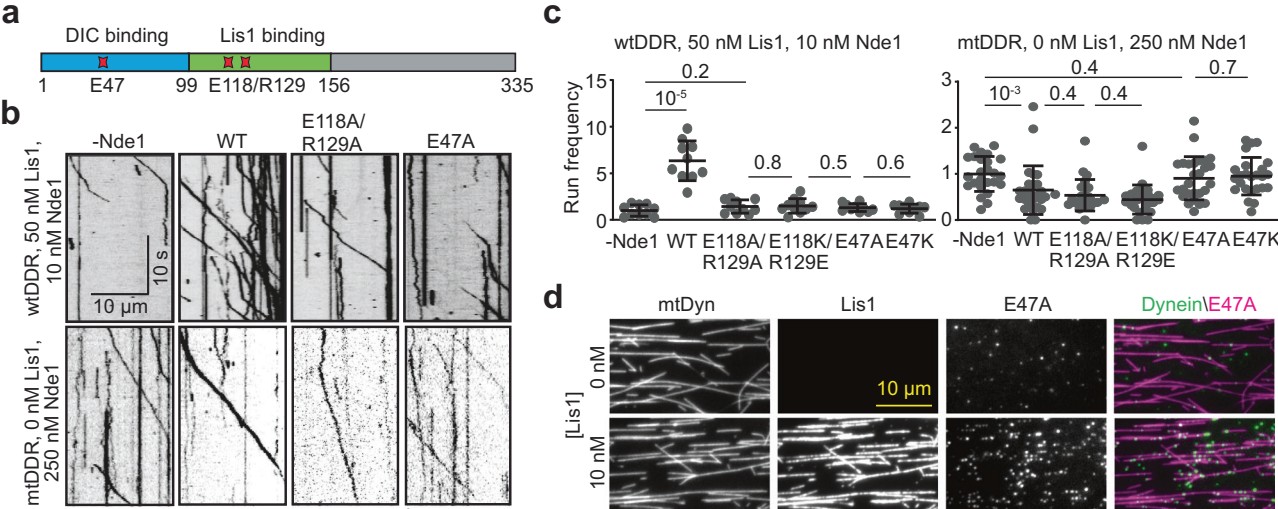

**Fig. 5 | Distinct roles of dynein and Lis1 binding of Nde1 in dynein regulation.** **a** Schematics show critical residues that facilitate Nde1 binding to Lis1 (E118 and R129) and dynein (E47). **b** Representative kymographs show wtDDR motility with or without 10 nM WT or mutant Nde1 with 50 nM Lis1 (Top) and mtDDR motility with 250 nM of WT or mutant Nde1 in the absence of Lis1 (Bottom). **c** Normalized run frequency distribution of wtDDR with or without 10 nM WT or mutant Nde1 with Lis1 (Left) and mtDDR with 250 nM of WT or mutant Nde1 in the absence of Lis1

(Right). From left to right, $n = 10, 10, 10, 10, 10, 10, 25, 27, 25, 26, 27$, and 25 microtubules. Results were normalized to the 0 nM Nde1 condition (-Nde1). The center line and whiskers represent the mean and s.d., respectively. $P$ values are calculated from a two-tailed $t$ test. **d** Colocalization of 25 nM Nde1$^{E47A}$ to dynein on surface-immobilized microtubules in the presence and absence of Lis1 (Two independent experiments). Source data are provided as a Source Data file.

immediately release from dynein while the other half comigrated with DDR on microtubules (Fig. 4k, l). These results indicate that Nde1 can remain associated with motile DDR complexes through Lis1 after the initiation of dynein motility. We note that this indirect association may be transient and Nde1 may slowly dissociate from motile complexes as we observe only 10% of DDR complexes to comigrate with Nde1 when Lis1 and Nde1 were preincubated with DDR components before testing the motility (Fig. 4c, d).

### Both dynein and Lis1 binding of Nde1 are required for dynein activation

To reveal how dynein binding and Lis1 binding of Nde1 contribute to the regulation of dynein motility, we generated Nde1 mutants that either cannot bind to DIC or Lis1. The point mutation to E47 of Nde1 has been shown to inhibit its binding to DIC[38]. Based on previous mutagenesis studies of Ndel1[30,38] and our AlphaFold2 model, E118 and R129 facilitate Nde1 binding to Lis1 (Fig. 2a). We generated both alanine substitutions and charge reversal mutations and showed that these mutants selectively disrupt Lis1 and DIC binding of Nde1 (Fig. 5a, Supplementary Fig. 6a). Mass photometry assays confirmed that the mutations did not disrupt dimerization of Nde1 (Supplementary Fig. 6b). The Lis1 binding mutants (Nde1$^{E118A/R129A}$ and Nde1$^{E118K/R129E}$) did not form a complex with Lis1, but colocalized with dynein in single-molecule colocalization assays (Supplementary Fig. 6c–e). Similarly, the DIC binding mutants (Nde1$^{E47A}$ and Nde1$^{E47K}$) did not colocalize to dynein (Supplementary Fig. 6c, d), but maintained their association with Lis1 (Supplementary Fig. 6e). Previous studies reported that the C terminal of Ndel1 interacts with DHC[36,58], but in vitro studies did not find evidence for Nde1 to bind DHC[31]. Because we observed a point mutant on the N-terminal coiled-coil of Nde1 to fully disrupt dynein binding, our results confirm that the C terminus of Nde1 does not bind dynein.

Motility assays showed that, unlike wild-type (WT) Nde1, none of the Nde1 mutants (10 nM) enhanced the run frequency of wtDDR in the presence of Lis1 (Fig. 5b, c, Supplementary Fig. 6f, and Supplementary Movie 4), showing that both Lis1 and DIC binding of Nde1 are required for Lis1-mediated activation of dynein. We then tested the inhibitive

effect of 250 nM Nde1 mutants in the absence of Lis1. Both WT and Lis1-binding mutants of Nde1 decreased the frequency of mtDDR runs by ~50% compared to the no Nde1 condition (Fig. 5b, c, and Supplementary Fig. 6f). However, DIC-binding mutants of Nde1 did not affect the run frequency (Fig. 5b, c, and Supplementary Fig. 6f), indicating that excess Nde1 competes with dynactin for DIC binding and negatively regulates the formation of the DDR complex.

We also used a DIC binding mutant (Nde1$^{E47A}$) to test whether a Lis1 dimer can simultaneously interact with dynein and Nde1 despite their overlapping binding sites on Lis1. Because this mutant cannot directly bind to dynein, it can be recruited to dynein only via Lis1 if one β-propeller can bind Nde1 while the other β-propeller binds dynein. Consistent with the inability of Nde1$^{E47A}$ to interact with DIC, we did not see colocalization of Nde1$^{E47A}$ to surface-immobilized dynein in the absence of Lis1. The addition of Lis1 resulted in colocalization of Nde1$^{E47A}$ to dynein (Fig. 5d), supporting the idea that a Lis1 dimer can form a ternary complex with Nde1 and dynein with one β-propeller interacting with Nde1 and the other interacting with dynein.

### Nde1 may compete against the sequestering of Lis1 by PAF-AH1B

In addition to its regulatory role in the dynein activation pathway, Lis1 also serves as the noncatalytic β subunit of the PAF-AH1B complex in vertebrates[59] and can modulate PAF-AH1B enzyme activity in vitro[60]. It remains mysterious whether Lis1's roles in these two regulatory pathways are coupled together, but the removal of the α subunit of PAF-AH1B did not result in defects in neurodevelopment[61], suggesting that the Lissencephaly phenotype is distinct from Lis1's role in the PAF-AH1B pathway. Overexpression of catalytic α1 or α2 subunits of PAF-AH1B has been shown to result in the inactivation of dynein-driven processes, whereas further overexpression of Lis1 or Nde1 restores dynein function[62]. Therefore, Nde1 and α1/α2 appear to compete for recruiting available Lis1 in the cytosol.

Nde1 has been shown to compete with α2 to bind Lis1 in vitro[14], but it remained unclear how Nde1 and α2 regulate the Lis1-mediated activation of dynein. To address this question, we assayed the competitive binding of α2 and Nde1 to Lis1 and determined how α2 affects the regulation of DDR motility by Lis1/Nde1. Previously reported

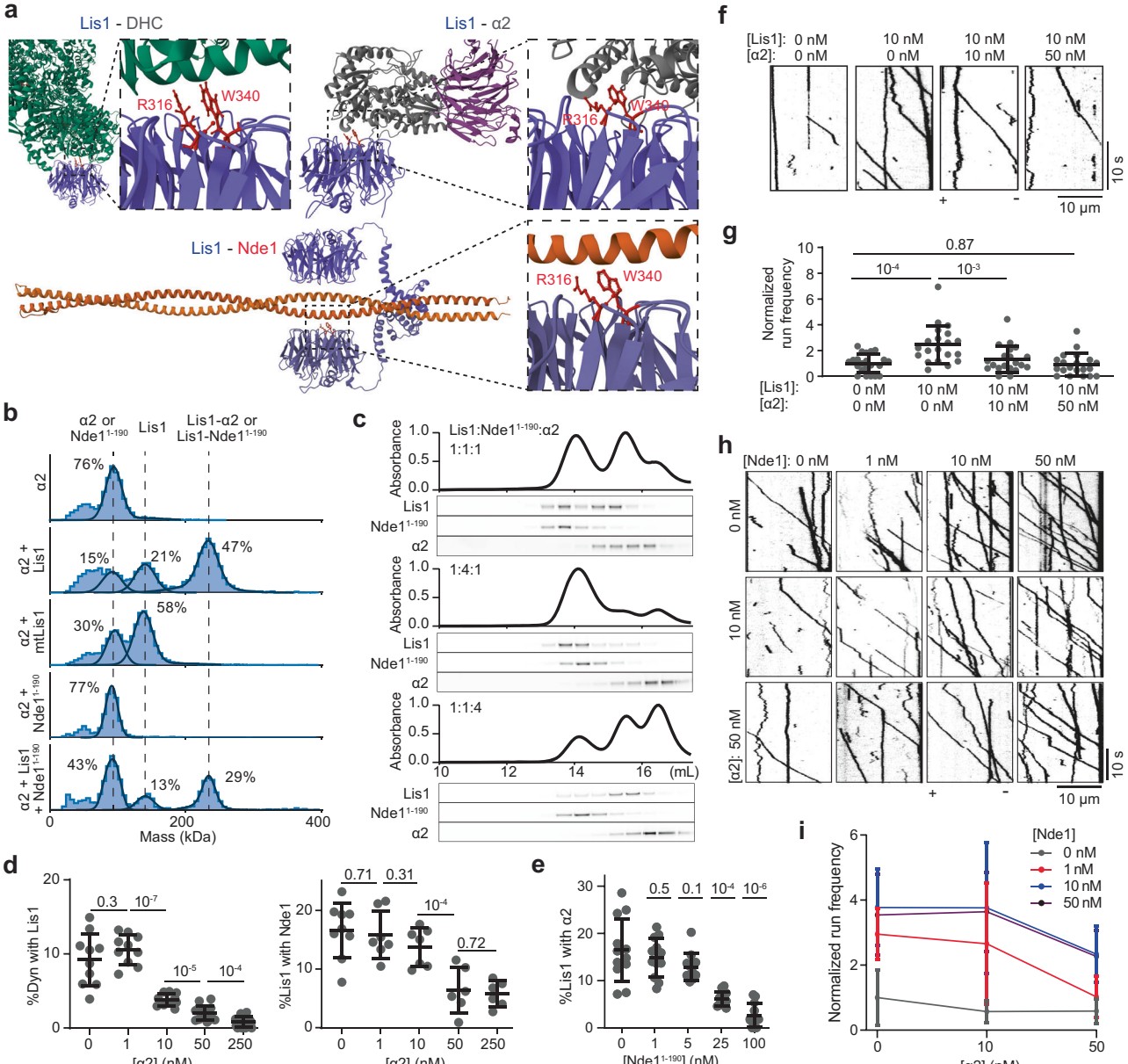

**Fig. 6 | Lis1 binding to dynein is blocked by the Lis1-α2 interaction in the absence of Nde1. a** Structures of Lis1-DHC (PDB ID: 8DYV[63]), Lis1-α2 (PDB ID: 1VYH[14]), and Lis1-Nde1[1–190] (predicted by AlphaFold2). Lis1 residues R316 and W340 that facilitate binding to dynein are also critical for binding to Nde1 and α2. **b** Mass photometry reveals that α2 forms a complex with Lis1, but not with mtLis1, and does not simultaneously bind to Lis1 and Nde1[1–190]. A multiple Gaussian fit predicts the average mass and percentage of each population (solid curves, Supplementary Table 2). **c** Normalized absorbance (top) and fluorescence picture of a denaturing gel (bottom) of the mixtures of Lis1-LD655, Nde1[1–190]-LD555, and α-Alexa488 eluted from a size exclusion column. **d** (Left) Ratios of surface-immobilized dynein colocalized with 1 nM Lis1 under increasing concentrations of α2. (Right) Ratios of surface-immobilized Lis1 colocalized with 25 nM Nde1 under increasing concentrations of α2. From left to right, $n = 10, 10, 10, 14$, and 19 (Left) and 9, 7, 7, 6, and

6 (Right) imaging areas (40 μm × 40 μm) with minimum 100 dyneins (Left) and Lis1 (Right). **e** Ratios of surface-immobilized Lis1 colocalized with 10 nM α2 under increasing concentrations of Nde1[1–190]. For each condition, $n = 12$ imaging areas (40 μm × 40 μm) with minimum 40 Lis1. **f** Kymographs show wtDDR motility under 0–50 nM α2 and 0–10 nM Lis1 without Nde1. **g** The run frequency of wtDDR under 0–50 nM α2 and 0–100 nM Lis1 without Nde1. Results were normalized to no Lis1 and α2 condition. From left to right, $n = 22, 20, 20$, and 20 microtubules. **h** Kymographs show wtDDR motility under different concentrations of α2 and Nde1 in 10 nM Lis1. **i** The run frequency of wtDDR under different concentrations of α2 and Nde1 in 10 nM Lis1 (mean ± s.d.; $n = 20$ microtubules for each condition). Results were normalized to the 10 nM Lis1 condition. In (**d**, **e**, **g**), the center line and whiskers represent the mean and s.d., respectively. *P* values are calculated from a two-tailed *t* test. Source data are provided as a Source Data file.

structures of DHC-Lis1[63] and α2-Lis1[14] and our AlphaFold2 model of Nde1-Lis1 show that dynein, α2, and Nde1 share the same binding site on Lis1 (Fig. 6a). Consistent with this model, mass photometry assays showed that α2 forms a complex with Lis1 but does not bind mtLis1 that interacts with neither dynein nor Nde1[1–190] (Fig. 6b). In addition, Lis1 cannot simultaneously interact with α2 and Nde1[1–190] (Fig. 6b). Because Nde1[1–190] and α2 constructs we used had similar masses, which

of these proteins formed a complex with Lis1 could not be distinguished from mass measurements (Fig. 6b). We used size exclusion experiments, which enabled us to resolve the peaks of Nde1-Lis1 and α2-Lis1 due to the elongated shape Nde1[1–190] (Fig. 6c and Supplementary Fig. 7). Nde1 and α2 formed a complex with Lis1 at equal ratios when mixed at equal concentrations. Increasing the relative concentration of Nde1 favored the formation of Nde1-Lis1, and similarly,

increasing the concentration of α2 favors the formation of α2-Lis1 (Fig. 6c), suggesting that Nde1 and α2 have similar affinities to bind Lis1[14].

Single-molecule colocalization assays showed that Lis1 exhibits reduced colocalization to both dynein and Nde1 under the increasing concentration of α2 (Fig. 6d). Similarly, increasing the concentration of Nde1 reduces the colocalization of α2 to surface-bound Lis1 (Fig. 6e). Single-molecule motility assays showed that, in the absence of Nde1 and Lis1, α2 does not directly alter the run frequencies of wtDDR or mtDDR (Supplementary Fig. 8). However, α2 addition lowers the wtDDR run frequency in a dose-dependent manner in the presence of Lis1 (Fig. 6f, g), suggesting that α2 downregulates Lis1 function by preventing its interaction with dynein. We also observed dose-dependent inhibition of wtDDR by α2 and activation by Nde1 over a wide range of concentrations (0–50 nM), suggesting that Nde1 rescues Lis1 from inhibition by α2 and tethers it to dynein (Fig. 6h, i). Consistent with this conclusion, premixing Lis1 and α2 during DDR assembly decreases the run frequency compared to the Lis1-only condition, but this can be rescued by Nde1 addition during complex assembly (Supplementary Fig. 9). These results suggest that Nde1 is needed to compete against sequestering of Lis1 by α1/α2 subunits of PAF-AH1B to facilitate dynein motility.

## Discussion

In this study, we investigated how Nde1 regulates dynein by reconstituting the motility of mammalian DDA complexes in the presence and absence of human Lis1 and Nde1. We showed that Nde1 recruits Lis1 to dynein. While Lis1 alone increases the frequency of DDA runs by ~3-fold, Nde1 and Lis1 together increase the run frequency up to 16-fold. This significant increase in the run frequency of wtDyn was similar to that of mtDyn, which effectively forms a complex with dynactin and a cargo adaptor in the absence of other regulatory factors[6]. Therefore, both Nde1 and Lis1 are needed to efficiently activate autoinhibited dynein.

The assembly of processive DDA complexes requires the opening of phi dynein[6], which is primarily driven by Lis1 binding to the AAA+ ring[11,16–19]. Based on our results and previous reports, we provide a mechanistic explanation for why another cellular factor (Nde1/Ndel1) is needed if Lis1 can bind and activate dynein on its own. We propose that Nde1/Ndel1 has two major roles in the dynein activation pathway (Fig. 7). First, Nde1/Ndel1 may compete against α1/α2 subunits of PAF-AH1B for Lis1. Because haploinsufficiency of Lis1 is sufficient to disrupt dynein function in cells[40] and cause disease[11], Lis1 is likely to be present in limited amounts in the cell and Nde1/Ndel1 and α1/α2 may compete

for cytosolic Lis1 for their regulatory roles in dynein and PAF-AH1B pathways, respectively. In this case, the absence of Nde1/Ndel1 might result in the sequestration of Lis1 to PAF-AH1B and, therefore, preventing the recruitment of Lis1 to dynein[62]. Second, Lis1 may not efficiently interact with autoinhibited dynein without Nde1/Ndel1, because the Lis1 binding site at the AAA+ ring appears inaccessible in the phi conformation[11,17,18]. While the Nde1/Ndel1 binding site on DIC has not been observed in available structures of the dynein complex[64], our results indicate that Nde1/Ndel1 can access DIC in the phi conformation and tether Lis1 to autoinhibited dynein.

We propose a mechanistic model of how Nde1 functions together with Lis1 in the dynein activation pathway (Fig. 7). Key features of this model are the overlapping binding of Nde1, α2, and dynein on the β-propeller domain of Lis1[14,15] and overlapping binding of Nde1 and the p150[Glued] subunit of dynactin on DIC[32,39,40,46,47]. Nde1 competes against α1/α2 subunits of PAF-AH1B and recruit about one-half of cytosolic Lis1 for the dynein pathway[62]. Nde1 tethers Lis1 to phi dynein by binding to DIC. Tethering of Lis1 increases its local concentration and may enable more efficient binding to the AAA+ ring once phi dynein transiently switches to the open conformation. Alternatively, Lis1 may directly bind and open phi dynein by wedging between the two AAA+ rings of a dynein dimer[65], and tethering of Lis1 to dynein may stabilize this intermediate step in dynein activation.

Because dynein and Nde1 bind to the same site on the β-propeller domain of Lis1, it remains unclear how Lis1 dissociates from Nde1 and binds to dynein. We reason that Lis1 may simultaneously interact with Nde1 and dynein via its two β propeller domains. Consistent with this view, we showed that a dynein-binding mutant of Nde1 can be recruited to dynein via Lis1 (Fig. 5d). Structural studies also observed that Lis1 binds to dynein primarily by one of its β-propeller domains near the AAA3 site, whereas the other β-propeller is either unbound or bound to dynein's stalk using a different interaction surface[15,66]. Therefore, one β-propeller may interact with Nde1 and the other β-propeller dissociates from Nde1 and binds to the AAA+ ring as Lis1 is being transferred from Nde1 to dynein.

After opening the phi conformation, dynein forms an active complex with dynactin and a cargo adaptor[6]. The interaction between the coiled-coil arm of the p150[Glued] subunit of dynactin and the N-terminus of DIC was shown to be critical for dynactin binding to dynein[67]. Because p150[Glued] and Nde1 have overlapping binding sites on DIC, Nde1 binding to DIC is incompatible with DDA assembly. Dynactin efficiently competes against Nde1 for DIC and releases Nde1 from the complex before the initiation of processive motility (Fig. 7). It remains to be determined why Nde1 and dynactin do not inhibit each other in

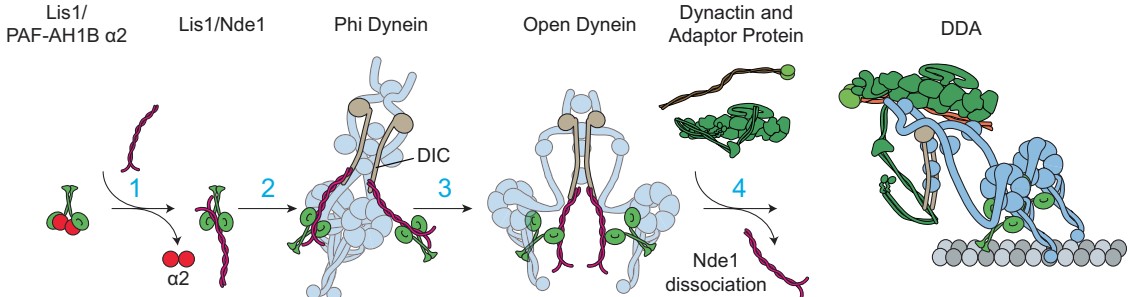

**Fig. 7 | Proposed model for the activation of the dynein transport machinery by Lis1/Nde1.** Nde1, dynein, and the α2 subunit of PAF-AH1B have overlapping binding sites on the β-propeller domain of Lis1. Without Nde1, Lis1 binding to dynein may be blocked by Lis1-α2 interaction. (1) Nde1 competes against α2 to bind Lis1. (2) Dynein forms an autoinhibited phi conformation through the interactions between its motor domains. While the Lis1 binding site at the AAA+ ring is inaccessible in the phi conformation, Nde1 binds to DIC and positions Lis1 near its dynein binding site. (3) Transient opening of dynein enables Nde1-tethered Lis1 to bind the dynein motor domain with one of its β propeller domains. Lis1 binding stabilizes the open conformation of dynein because the Lis1-bound dynein is sterically incompatible with the phi conformation. (4) Open dynein binds to dynactin and a cargo adaptor and switches to the parallel conformation. Upon binding to open dynein, dynactin interacts with DIC and releases Nde1 from this site. Subsequently, Lis1 also dissociates from dynein during or after the initiation of processive motility. While binding of two Lis1/Nde1 per dynein has been shown in this model, single Lis1/Nde1 binding may also be sufficient to activate DDA assembly.

cells despite their overlapping binding sites on DIC. One possible explanation is that Nde1 and dynactin perform their regulatory roles in two distinct conformations of dynein. According to this scheme, dynactin does not prevent recruitment of Nde1 to phi dynein because it does not strongly interact with dynein in this conformation[6]. The p150[Glued] arm of isolated dynactin is observed to be folded back and docked onto the pointed end of dynactin, blocking the binding site of cargo adaptors[68]. Docking of the p150[Glued] arm may also prevent its interaction with DIC. Recruitment of open dynein and a cargo adaptor to dynactin results in undocking of the p150[Glued] arm[68], which may also activate p150[Glued]'s interaction with DIC. As a result, dynactin may trigger the dissociation of Nde1 from DIC after the opening of phi dynein. Predictions of this model and the order of events that leads to the activation of the dynein transport machinery need to be tested by future structural and biophysical studies.

Our model also provides a mechanistic explanation for conflicting observations made for Nde1 and Lis1 in vivo and in vitro. While cell-based assays showed that Nde1 is crucial for the recruitment of Lis1 to activate dynein[27,36,40,58,69], several in vitro studies reported inhibition of dynein by Nde1[31,48]. We showed that Nde1 can serve as both the positive and negative regulator of dynein under different concentration regimes. A low concentration of Nde1 is sufficient to tether Lis1 to dynein and promote dynein motility. We propose that this concentration regime represents the physiological function of Nde1 in the dynein activation pathway. Consistent with this view, cellular studies have shown that Ndel1 depletion can be rescued by overexpression of Lis1, because increasing the cellular concentration of Lis1 may be sufficient to stimulate Lis1 binding to dynein and DDA assembly even in the absence of Nde1[38]. However, Ndel1 overexpression cannot rescue Lis1 deletion[38], because Nde1 is incapable of activating dynein without Lis1. We also observed that a very high concentration of Nde1 could play an inhibitory role by competing against DIC-dynactin interaction in vitro, consistent with excess Nde1 to cause deleterious effects on dynein function in cells[22,38,43,44]. We propose that the physiological concentration of Nde1 is lower than this inhibitory regime and dynactin can efficiently displace Nde1 from dynein during the assembly of active complexes.

Our results are also largely consistent with two concurrent in vitro studies, which reported that Ndel1 tethers Lis1 to dynein and competes against p150[Glued] binding to dynactin[67,70]. While Garrott et al.[70] also reported that Ndel1 disfavors the formation of active DDA complexes in the absence of Lis1, they did not observe the activating role of Ndel1 in dynein motility when added together with Lis1. This discrepancy could be related to the differences between the two paralogs (Nde1 vs. Ndel1) or activating adaptors (BiCDR1 vs. BicD2) used in our studies. Future studies are required to distinguish between similar but potentially distinct cellular roles of Nde1 and Ndel1, as deletion of Nde1 causes microcephaly while the loss of Ndel1 is usually fatal[29]. We also note that Nde1/Ndel1 have been reported to interact with other cellular factors and undergo posttranslational modifications[29]. While most of the identified phosphorylation sites are located in the C-terminus, phosphorylation of Nde1 at T131 (equivalent to T132 of Ndel1) has been shown to reduce its ability to interact with Lis1[71]. The in vitro reconstitution assay we developed could serve as a platform to test how these interaction partners and post-translational modifications regulate the proposed roles of Nde1/Ndel1 in the dynein activation pathway.

## Methods

### Protein expression and purification

The plasmids with the pOmniBac backbone were transformed into DH10Bac competent cells and plated onto Bacmid plates with BluoGal at 37 °C for 2 days. A white colony was selected and grown in 2X-YT media overnight. Bacmid plasmids were purified and transfected onto adherent SF9 cells. The transfected SF9 cells were incubated at 27 °C for 3 days to grow the p1 virus. Then 2 mL of p1 virus was added to 50 mL suspended SF9 cell culture and incubated at 27 °C in a shaking incubator for 3 days to obtain the p2 virus. Then p2 virus was collected by centrifuging at 4000 g for 10 min and stored at 4 °C in the dark for long-term use.

For protein expression, 1 L suspended SF9 cell culture was infected by the p2 virus with the multiplicity of infection (MOI) at 3 and incubated for 3 days. Cells were collected by centrifuging at 4000 g for 10 min. Then the pellets were either immediately lysed for protein purification or snap-frozen in liquid nitrogen and stored at −80 °C.

To extract proteins, SF9 pellets were re-suspended into the lysis buffer (25 mM HEPES pH 7.4, 150 mM KAc, 1 mM MgCl₂, 1 mM EGTA, 1 mM DTT, 0.1 mM ATP, 20 mM PMSF, and 10 Roche protease inhibitor tablets per L) and lysed by a dounce homogenizer. Lysate was cleared at 150,000 g for 30 min in a Ti70 rotor (Beckman Coulter) and the supernatant was incubated with 1 mL IgG Sepharose beads (GE Healthcare) for 1 h at 4 °C. The beads were collected and washed with lysis buffer and then with the TEV buffer (25 mM HEPES pH 7.4, 150 mM KAc, 1 mM MgCl₂, 1 mM EGTA, 1 mM DTT, 0.1 mM ATP). To elute the proteins from the beads, 0.1 mg/mL TEV protease was added and rolled at a nutator for 1 h at room temperature. Proteins were separated from the beads using a 0.45 μm pore-sized centrifugation filter (Amicon Ultrafree MC) and concentrated with 50 K molecular weight cut-off (MWCO) concentrators (Amicon).

For fluorescent labeling, proteins were incubated with 4-fold excess dye derivatized with either benzylguanine (BG, for SNAP labeling) or coenzyme A (CoA, for ybbR labeling) at 37 °C for 1 h. Dynein was labeled at room temperature. 5 μM SFP enzyme was added to catalyze ybbR labeling with CoA. The probability, $p$ of each SNAP-tagged monomer with benzyl guanine derivatized dyes was 0.65. The probability of a SNAP-Lis1 dimer to be labeled with at least one dye (calculated as $2p - p^2$) was 0.88. The labeling efficiency of the ybbR tag was 0.50 and the probability of dimeric proteins labeled with at least one dye was 0.75. Labeled proteins in TEV buffer were eluted from a size exclusion column to remove the free dye and other impurities. Dynein was eluted from the TSKgel G4000SWXL column (Tosoh), BicDR1 was eluted from the Superose 6 10/300 GL column (Cytiva), whereas Lis1, Nde1, and α2 were eluted from the Superdex 200 Increase 10/300 GL column (Cytiva) (Supplementary Fig. 10).

Dynactin was purified from pig brains using SP Sepharose Fast Flow and MonoQ ion exchange columns (Cytiva) and the TSKgel G4000SWXL size exclusion column (Tosoh), as previously described[72]. Dynactin was labeled with LD555 derivatized with NHS and excess dye was removed by passing the dynactin solution through a desalting column (Zeba).

### Microscopy

The fluorescent imaging was performed with a custom-built multicolor objective-type TIRF microscope equipped with a Nikon Ti-E microscope body, a 100X magnification 1.49 N.A. apochromatic oil-immersion objective (Nikon) together with a Perfect Focus System. The fluorescence signal was detected using an electron-multiplied charge-coupled device camera (Andor, Ixon EM⁺, 512 × 512 pixels). The effective camera pixel size after magnification was 160 nm. Alexa488/GFP/mNeonGreen, LD555, and LD655 probes were excited using 488 nm, 561 nm, and 633 nm laser beams (Coherent) coupled to a single mode fiber (Oz Optics), and their emission was filtered using 525/40, 585/40, and 697/75 bandpass filters (Semrock), respectively. The microscope was controlled by MicroManager 1.4.

### Preparation of flow chambers

Glass coverslips were coated with polyethylene glycol (PEG) to reduce the nonspecific binding of proteins[73]. Plain glass coverslips were cleaned with water, acetone, and water by sonication for 10 min sequentially, and then sonicated in a 1 M KOH using a bath sonicator

for 40 min. The coverslips were then rinsed with water, incubated in 3-Aminopropyltriethoxysilane in acetate and methanol for 10 min with 1-min sonication between successive steps, cleaned with methanol, and air-dried. 30 μl of 25% biotin-PEG-succinimidyl valerate in a NaHCO$_3$ buffer (pH 7.4) was sandwiched by two pieces of coverslips at 4 °C overnight. The coverslips were cleaned with water, air-dried, vacuum sealed, and kept at −20 °C for long-term storage. Flow chambers were built by sandwiching a double-sided tape with a PEG-coated coverslip and a glass slide. To flow a solution into a flow chamber while recording DDR motility in real time, two holes were drilled at each end of the chamber on the glass slides.

## Single-molecule motility assays

The flow chambers were incubated with 5 mg ml$^{-1}$ streptavidin for 2 min and washed with MB buffer (30 mM HEPES pH 7.0, 5 mM MgSO$_4$, 1 mM EGTA, 1 mg ml$^{-1}$ casein, 0.5% pluronic acid, 0.5 mM DTT, and 1 μM Taxol). The chamber was then incubated with biotinylated microtubules for 2 min and washed with MB buffer. Proteins were diluted and mixed into desired concentrations in MB buffer. For DDR assembly, 10 nM dynein, 150 nM dynactin, and 50 nM BicDR1 were incubated on ice for 25 min, then diluted 10-fold into imaging buffer (MB buffer supplemented with 0.1 mg ml$^{-1}$ glucose oxidase, 0.02 mg ml$^{-1}$ catalase, 0.8% D-glucose, and 2 mM ATP), and introduced to the flow chamber. Motility was recorded for 5 min.

## Single-molecule colocalization assays

Single-molecule colocalization assays were performed by labeling the SNAP-tagged "bait" protein (either dynein or Lis1) with equal concentrations of Alexa488-BG and biotin-BG. Biotin-PEG coated flow chamber was incubated with 5 mg ml$^{-1}$ streptavidin for 2 min and washed with MB buffer. Then the bait protein was diluted to 0.2 nM in MB buffer and incubated in the chamber for 1 min. Unbound protein was removed by washing the chamber with MB buffer. The "prey" proteins were labeled with LD655, diluted in MB buffer, flowed into the chamber, and incubated for 10 min. The fluorescence signals of Alexa488 and LD655 were recorded without washing away the unbound prey proteins in the solution. The percentage of colocalization was typically lower than 50% due to the incomplete labeling of the proteins and the necessity to use low concentrations of labeled proteins to remain within the single molecule detection limit (less than 1 spot per μm$^2$).

## Co-immunoprecipitation

GFP-tagged proteins were labeled with Alexa-488 for Typhoon imaging in Coomassie stained denaturing gels. GFP-Trap beads (Chromotek) were incubated with MB buffer supplemented with 5 mg ml$^{-1}$ BSA overnight at 4 °C to minimize non-specific binding. Proteins were mixed with Co-IP buffer (MB buffer supplemented with 5 mg ml$^{-1}$ BSA and 150 mM NaCl) and diluted into desired concentrations. 20% of the protein mix was separated into another tube as "Input". The remaining 80% was incubated with GFP-Trap beads and incubated for 1 h on ice. The beads were then washed with Co-IP buffer and centrifuged at 2000 g for 3 min three times to remove unbound protein in the supernatant. Input and beads were run in a denaturing gel (NuPAGE, Thermofisher). The gel was imaged using GF Typhoon FLA 9500 A (GE Healthcare) to detect the fluorescence signal of the labeled proteins that eluted with the beads.

The validation data of GFP-TRAP Agarose beads is available on the product website (https://www.ptglab.com/products/GFP-Trap-Agarose-gta.htm).

## Mass photometry

High-precision coverslips (Azer Scientific) were cleaned with isopropanol and water alternatively 3 times in a bath sonicator and air-dried. The gasket was cleaned with isopropanol and water alternatively

3 times without sonication, air-dried, and placed onto a clean coverslip. 14 μL of mass photometry buffer (30 mM HEPES pH 7.4, 5 mM MgSO$_4$, 1 mM EGTA, and 10% glycerol) was loaded onto a well for the autofocus. The protein sample was diluted to 5–20 nM in mass photometry buffer and added to the coverslips. Protein contrast count was collected with a TwoMP mass photometer of Refeyn 2 with two technical replicates. Mass measurements of the instrument were calibrated using the standard mix (conalbumin, aldolase, and thyroglobulin). Mass photometry profiles were fitted to multiple skewed Gaussian peaks and their mean, standard deviation, and percentages were calculated using the DiscoverMP software (Refeyn).

## Analytical size exclusion column (SEC)

To perform analytical SEC, Lis1, Nde1, and α2 were labeled with LD655, LD555, and Alexa488, respectively. The proteins were mixed at the desired ratios and incubated for 10 min at 4 °C. The mixture was then loaded onto the Superose 6 10/300 Increase column (Cytiva). The elution was collected and run on a denaturing gel. The gel was imaged under Typhoon Imager.

## Data analysis

Single-molecule motility of DDR was recorded for 500 frames per imaging area and analyzed using the z-stack function of FIJI to determine the orientation of unlabeled microtubules on the surface. Microtubule tracks longer than 10 μm have been included in data analysis. The run frequency was calculated by observing the number of processive BicDR1 on each microtubule divided by the length of the microtubule and the duration of data collection using a custom-written MATLAB code. Velocity was calculated by detecting the start and the end of each processive run of BicDR1. The fraction of single molecule colocalization was calculated by dividing the number of LD655 spots that colocalize with Alexa488 by the total number of Alexa 488 spots on the 40 μm × 40 μm imaging area. Colocalization was defined as the maximum 300 nm distance between the peaks of diffraction-limited spots of Alexa488 and LD6555 dyes. The localization of dyes was detected by a modified version of FIESTA[74] (YFIESTA, available on https://github.com/Yildiz-Lab/YFIESTA).

## AlphaFold2 protein structure prediction

AlphaFold2 structure prediction for Nde1$^{1-190}$-Lis1 and Nde1 was performed on Google ColabFold with AlphaFold2_mmseqs2 version (available on https://github.com/sokrypton/ColabFold) using default settings. The structure of Nde1-Lis1 was generated with structural alignment of Nde1$^{1-190}$-Lis1 and Nde1 structure prediction based on the amino acids 1-190 of Nde1.

The protein structures used in Fig. 6a are
Lis1-α2 (1VYH).
Lis1-DHC (8DYV).

The images and movies of the structural models were created on the RCSB website.

## Statistical analysis

The p-values were calculated by the two-tailed t test in Prism and Origin. CDFs were calculated in MATLAB.

## Reporting summary

Further information on research design is available in the Nature Portfolio Reporting Summary linked to this article.

# Data availability

A Reporting Summary for this article is available. The data supporting the findings of this study are available within the article and in its Supplementary Information and Source Data file. The constructs that express wild-type and mutant versions of Lis1, Nde1, and α2 will be deposited to AddGene. Raw microscopy data will be made available by

the corresponding author upon request. Source data are provided with this paper.

## Code availability
The custom code used to analyze experimental data is uploaded to the Yildiz Lab code repository [https://github.com/Yildiz-Lab] and GitHub [https://github.com/Yildiz-Lab/YFIESTA].

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

## Acknowledgements

The authors are grateful to the members of the Yildiz laboratory for helpful discussions and carefully reading the manuscript, Andrew P. Carter for providing dynein expression plasmids, and Eva Nogales and Juan P. Bertoldi for mass photometry. This work was funded by grants from the NIH (GM136414), and NSF (MCB-1055017, MCB-1617028) to A.Y.

## Author contributions

Y.Z. and A.Y. conceived the study and designed the experiments. Y.Z. prepared the constructs and isolated the proteins. Y.Z. and S.O. labeled the proteins with fluorescent dyes and performed the motility and Co-IP experiments. Y.Z. and A.Y. wrote the manuscript, and all authors read and commented on the manuscript.

## Competing interests

The authors declare no competing interests.
