## [Peer Review File · Nature Communications]

Nde1 Promotes Lis1-Mediated Activation of DyneinREVIEWER COMMENTS

Reviewer #1 (Remarks to the Author):

Comment

The motility of the minus-end directed microtubule motor dynein is regulated by a pair of dynein-associated proteins, Lis1 and Nde1/Ndel1 (Nde1), which are both interactors of dynein and of one another. While several studies have recently clarified the role of Lis1 as an activator of dynein motility, the role of Nde1 has remained largely obscure.

Zhao and coworkers use *in vitro* TIRF assays, mass photometry, and other single-molecule techniques to dissect how Nde1, Lis1, and the Lis1 inhibitory partner PAFAH- α 2 interact to regulate dynein motility. Overall, the data presented in this paper is of high quality and the conclusions are well evidenced, and this paper will be of significant interest to a broad audience interested in the structure and function of molecular motors.

The mass photometry data presented in this paper is of high quality, and the experiments are overall well executed, and showcase an interesting application of this novel technique. However, some changes to the presentation of these results would increase their clarity for readers.

Minor points

- All MP panels: The MP profile plots should include gaussian fits for the MP mass peaks.
- All MP panels: The theoretical and measured masses for all peaks should also be displayed, either as a table, or within the individual panels, to allow for easier interpretation.

Reviewer #2 (Remarks to the Author):

In this study by Zhao et al., the authors investigate the role of Nde1 in regulating cytoplasmic dynein. They propose that Nde1 competes with the α -subunit of PAFAH to release LIS-1 (PAFAH- β) thereby allowing LIS-1 to relieve the auto-inhibited state of the dynein motor domains. Consistent with this model, the ability of Nde1 to stimulate dynein motility is dependent on the presence of LIS-1. This activity for Nde1 is observed when Nde1 is provided at relatively low concentrations. However, when Nde1 is provided at much higher concentrations (1000 x), Nde1 inhibits dynein motility significantly. The authors propose that this reflects a competition for binding the dynein IC subunit between Nde1 and dynactin. Excess Nde1 prevents recruitment of dynactin and blocks the steps required for motility. This alternation between an activating and an inhibiting function for Nde1 is described as biphasic. This paper reports an interesting multi-step sequence for the activation of dynein. The authors using compelling experimental tools to tease the steps apart, and the *in vitro* motility assays are convincing. The make ample use of mutants which seems appropriate for the questions they are asking. The contributions of this manuscript will be exciting to a broad audience including investigators working on organelle transport. However, there are two overarching issues that deserve attention by the authors. 1) The concept of biphasic regulation of dynein reflects the differences in the role of Nde1 at two very different concentrations. The lower concentration appears to mirror the natural concentration of Nde1 in cells and probably represents the true function of Nde1 (dynein activation via LIS-1). The higher concentration (1000 x) seems highly artificial and probably does not reflect the normal situation in cells. Given that a competition for dynein IC binding between Nde1 and dynactin would require highly elevated Nde1 levels, it is not clear that this really happens in cells. A focus on the concentration used for these assays that matches the natural concentration of Nde1 in cells would improve the interpretation of this work significantly. This issue is important because the authors have chosen to focus on this concept of biphasic regulation.

2) The idea that Nde1 and dynactin compete for binding to the dynein ICs has many interesting implications. This concept has been reported previously by others. However, rather than using protein concentration changes as a mechanism of regulation, the authors might consider phosphorylation as the mechanism. This represents a much more common method of changing protein-protein interactions. Furthermore, many of the proteins under consideration in this study are known to be phosphorylation substrates. In some cases the specific sites of phosphorylation are known, and the impact of phosphorylation on protein-protein interactions has been reported. Phosphorylation site mutants are also available. It would be important to understand the relative contributions of phospho-regulation vs. biphasic regulation.

Reviewer #3 (Remarks to the Author):

Review of NCOMMS-23-13528

In this manuscript, Zhao et al sought to assess the assembly and motility of dynein-dynactin-adaptor DDR complex in the presence of Lis1, Nde1, and alpha2 using a TIRF imaging assay. While many of the protein-protein interactions have been previously described – such as between LIS1 and Nde1, between LIS1 and alpha2, between LIS1 and dynein motor domain, between Nde1 and dynein DIC, and between dynactin p150 and dynein DIC – the authors have done a nice job carrying out detailed characterizations, often by titrating one component against the other, to determine how each interaction affects the assembly of DDR complex and ultimately dynein motility. This has provided some new insights into how Nde1 and LIS1 cooperate to trigger the opening of autoinhibited phi dynein in a pathway leading to the assembly of an activated, processive DDR complex. Perhaps the most interesting result is that the authors have found Nde1 to serve as both a positive and negative regulator of dynein motility in a manner dependent on its concentration. They claim that, at low concentrations (e.g., 2 to 20 nM), Nde1 can recruit Lis1 to dynein to promote opening of phi dynein, whereas at high concentrations (e.g., 500 to 2000 nM), Nde1 can compete with dynactin p150 for dynein DIC interaction, thereby negatively regulating the formation of the DDR complex. This claim, if true, appears to offer an explanation to clarify the discrepancy in the literature (e.g., ref. 37, 51, and 30) on whether Nde1 enhances dynein motility or inhibits it. However, in this reviewer's opinion, the manuscript appears to lack important controls to support their claim, especially with regards to E47A and E47K mutations (DIC-binding mutations in Nde1), causing them to overinterpret their results. The mechanism by which excess Nde1 inhibits dynein activation pathway remains elusive. Importantly, although their results provided a possible explanation for why high overexpression of Nde1 is detrimental to dynein function in cells, the physiological significance of the proposed inhibitory effect involving excess Nde1 (i.e., involving excess Nde1 to compete with dynactin p150 for dynein DIC) remains unclear. Whether Nde1 has an inhibitory effect would depend on the physiological concentrations of Nde1 and dynactin p150 in the cell, and their respective affinity (dissociation constant) for dynein IC, which remains unknown. Additionally, the conceptual advance of this paper is diluted by the fact that previous studies have already demonstrated a role for Nde1/Nde1 in tethering Lis1 to dynein (e.g., ref. 37 and 51). Previous studies have also demonstrated that Nde1 competes with p150 subunit of dynactin for interaction with dynein DIC (ref. 31). Thus, in this reviewer's opinion, the manuscript is not appropriate for publication in its current state.

Major comments:

1. The author generated E47A and E47K mutations to selectively disrupt DIC-binding of Nde1. However, these Nde1 mutants at 20 nM (i.e., low concentration) failed to boost the wtDDR run frequency in the presence of 100 nM Lis1 (in Fig. 5c). They also failed to decrease the mtDDR run frequency at 500 nM (i.e., high concentration, Fig. 5e). Is it possible that these mutations resulted in a global change in the protein structure, causing unfolding and rendering the protein non-functional? The authors proceeded to conclude that both Lis1-binding and DIC-binding of Nde1 are required for Lis1-mediated activation of dynein, but the data do not fully support this conclusion. Evidence showing that these mutant proteins are folded properly should be provided.
2. If both Lis1-binding and DIC-binding are required for enhancing wtDDR run frequency in the presence of 100 nM Lis1 (at 20 nM Nde1, see page 7 and Fig. 5c), then what is the role for the interaction of Nde1 with the dynein motor domain in the pathway of forming the DDR complex? The authors seemed to mention that Nde1 can bind dynein through the motor domain (see page 6, second paragraph). A reference for this interaction should be provided. If Nde1 can bind dynein through DIC and/or the motor domain, then it is reasonable to think that motor-domain-binding by Nde1, instead of DIC-binding, could be involved in Step #2 (Nde1 mediating recruitment of Lis1 to dynein) of the model in Fig. 7.
3. Run frequency was calculated from the TIRF images and was used as a readout for dynein

activation pathway. While this seems like a good way to normalize the number of processive wtDDR on each microtubule for comparison under different conditions, why do the kymographs for the conditions with 2000 nM Nde1 (for example, those in Fig. 1d, 1f, and 2c) consistently show a dramatically smaller number of landing events compared to the conditions with lower concentrations of Nde1? Are these kymographs representative? Additionally, the authors should provide more clarity in the Methods section on how they determined the length of the microtubule when calculating the run frequency. Related to adding clarity, Fig. 1a did not specify how or if the surface-immobilized microtubules are fluorescently labeled.

4. The authors claimed that they observed only a small fraction of DDR complexes comigrated with Nde1 in three-color TIRF assays shown in Fig. 4c. However, the kymographs in this figure panel did not make sense. While the authors selectively pointed to tracks (with red arrows) showing comigration of Lis1-LD555 with a processive BicdR1-mNG, they seemed to have ignored tracks showing migration of Lis1-LD555 without BicdR1-mNG that are also visible on the kymographs. Can the authors explain why? The number of migrating Lis1-LD555 tracks without BicdR1-mNG relative to the number of migrating BicdR1-mNG tracks without Lis1-LD555 should be counted and reported. This might help uncover the noise that is inherent in this assay, and allow other labs to reproduce their observations.

5. In Fig. 4a, excess Nde1 was added before or after the assembly of the DDR complex. However, the label on the figure panel says "added during assembly". To avoid confusion, the authors should clarify the difference between during and before. Alternatively, the authors should use the same wording.

6. In Fig. 3f, why does increasing full-length Lis1 (from 1 to 2.5 to 10 nM) appear to decrease the association of Nde1(1-190) with dynein in single-molecule colocalization assay? This is inconsistent with Fig. 2f showing that increasing Nde1 would linearly increase the colocalization of Lis1 to surface-immobilized dynein in the same type of assay.

Minor comments:

7. Typo on page 3, last paragraph of Introduction, 5th line: "Nde1" should be "Nde1".

8. Error on page 9, middle paragraph, last sentence: "overexpression of Nde1...." is repeated twice.

9. Error in a sentence on page 7: "Lis1 has been initially identified as the noncatalytic β subunit of PAF-AH and the catalytically-active $\alpha 1$ and $\alpha 2$ subunits of PAF-AH to inactivate dynein by interacting with Lis1". This sentence implies that Lis1 has been initially identified as $\alpha 2$ subunit of PAF-AH.

We would like to thank the reviewers for carefully reviewing our manuscript. We were very glad to hear that the reviewers found this work of potential interest, but they also raised substantial concerns. These comments were very helpful and aided us in significantly improving the manuscript. In this revised manuscript, we

- 1) shifted the focus of the manuscript from “biphasic regulation” to the physiological role of Nde1 in the dynein activation pathway by revising the title, abstract, and main text. Briefly, we clarified our interpretation that activating role of Nde1 at low concentrations represents its physiological role, whereas its inhibitory role appears at concentrations much higher than the cellular concentration of Nde1, but these observations provide an explanation for some of the conflicting observations previously made in the field.
- 2) performed additional experiments to characterize the point mutants of Nde1.
- 3) added new experimental evidence to further validate our model.
- 4) responded to all other referee concerns about methodological details, clarifications, and textual changes.

A detailed explanation of these revisions can be found in our point-by-point response (highlighted in blue) to referee comments (highlighted in black) below.

REVIEWER COMMENTS

Reviewer #1

The motility of the minus-end directed microtubule motor dynein is regulated by a pair of dynein-associated proteins, Lis1 and Nde1/Ndel1 (Nde1), which are both interactors of dynein and of one another. While several studies have recently clarified the role of Lis1 as an activator of dynein motility, the role of Nde1 has remained largely obscure.

Zhao and coworkers use in vitro TIRF assays, mass photometry, and other single-molecule techniques to dissect how Nde1, Lis1, and the Lis1 inhibitory partner PFAH- α 2 interact to regulate dynein motility. Overall, the data presented in this paper is of high quality and the conclusions are well evidenced, and this paper will be of significant interest to a broad audience interested in the structure and function of molecular motors.

We thank the reviewer for their careful reading of our manuscript and for recognizing the impact of our work in the field.

The mass photometry data presented in this paper is of high quality, and the experiments are overall well executed, and showcase an interesting application of this novel technique. However, some changes to the presentation of these results would increase their clarity for readers.

Minor points

- All MP panels: The MP profile plots should include gaussian fits for the MP mass peaks.
- All MP panels: The theoretical and measured masses for all peaks should also be displayed, either as a table, or within the individual panels, to allow for easier interpretation.

We have fitted the MP profiles to Gaussian peaks and reported the percentage, mean, standard deviation, and predicted molecular weight of the fitted peaks in the main text figures and the

new Supplementary Table 2.

Reviewer #2

In this study by Zhao et al., the authors investigate the role of Nde1 in regulating cytoplasmic dynein. They propose that Nde1 competes with the α -subunit of PAFAH to release LIS-1 (PAFAH- β) thereby allowing LIS-1 to relieve the auto-inhibited state of the dynein motor domains. Consistent with this model, the ability of Nde1 to stimulate dynein motility is dependent on the presence of LIS-1. This activity for Nde1 is observed when Nde1 is provided at relatively low concentrations. However, when Nde1 is provided at much higher concentrations (1000 x), Nde1 inhibits dynein motility significantly. The authors propose that this reflects a competition for binding the dynein IC subunit between Nde1 and dynactin. Excess Nde1 prevents recruitment of dynactin and blocks the steps required for motility. This alternation between an activating and an inhibiting function for Nde1 is described as biphasic.

This paper reports an interesting multi-step sequence for the activation of dynein. The authors using compelling experimental tools to tease the steps apart, and the in vitro motility assays are convincing. They make ample use of mutants which seems appropriate for the questions they are asking. The contributions of this manuscript will be exciting to a broad audience including investigators working on organelle transport. However, there are two overarching issues that deserve attention by the authors.

1) The concept of biphasic regulation of dynein reflects the differences in the role of Nde1 at two very different concentrations. The lower concentration appears to mirror the natural concentration of Nde1 in cells and probably represents the true function of Nde1 (dynein activation via LIS-1). The higher concentration (1000 x) seems highly artificial and probably does not reflect the normal situation in cells. Given that a competition for dynein IC binding between Nde1 and dynactin would require highly elevated Nde1 levels, it is not clear that this really happens in cells. A focus on the concentration used for these assays that matches the natural concentration of Nde1 in cells would improve the interpretation of this work significantly. This issue is important because the authors have chosen to focus on this concept of biphasic regulation.

We agree with the reviewer's view that "The lower concentration appears to mirror the natural concentration of Nde1 in cells and probably represents the true function of Nde1" and that "The higher concentration seems highly artificial and probably does not reflect the normal situation in cells". In the original manuscript, we focused on the concept of biphasic regulation from the in vitro mechanistic point of view. These competing effects of Nde1 at different concentration regimes enabled us to provide an explanation for the conflicting results reported in previous in vitro and overexpression studies of Nde1/Nde1, and to potentially end the controversy in the regulatory roles of Nde1. In the revised manuscript, clarified our interpretation that activating role of Nde1 at low concentrations represents its physiological role, whereas its inhibitory role appears at concentrations much higher than the cellular concentration of Nde1, but these observations provide an explanation for some of the conflicting observations previously made in the field.

We clarified these points by 1) changing our title, 2) revising the abstract, and 3) substantially editing the Discussion. For example, in the Discussion of the revised manuscript, we wrote:

“Our model provides a mechanistic explanation for seemingly conflicting observations made for Nde1 and Lis1 in vivo and in vitro. While cell based-assays showed that Nde1 is crucial for the recruitment of Lis1 to activate dynein 26, 35, 38, 59, 66, several in vitro studies reported inhibition of dynein by Nde1^{30, 47}. We showed that Nde1 can serve as both the positive and negative regulator of dynein under different concentration regimes. A low concentration of Nde1 is sufficient to tether Lis1 to dynein, increasing its apparent affinity for dynein. We propose that this concentration regime represents the physiological function of Nde1 in the dynein activation pathway. Consistent with this view, cellular studies have shown that Nde1 depletion can be rescued by overexpression of Lis1, because increasing the cellular concentration of Lis1 may be sufficient to stimulate Lis1 binding to dynein and stimulate DDA assembly even in the absence of Nde1³⁶. However, Nde1 overexpression cannot rescue Lis1 deletion³⁶, because Nde1 is incapable of activating dynein without Lis1. We also observed that a very high concentration of Nde1 could play an inhibitory role by competing against DIC-dynactin interaction in vitro, providing an explanation for why excess Nde1 has deleterious effects on dynein function in cells 21, 36, 42, 43. We propose that the physiological concentration of Nde1 is lower than this inhibitory regime and dynactin can efficiently displace Nde1 from dynein during the assembly of active complexes.”

2) The idea that Nde1 and dynactin compete for binding to the dynein ICs has many interesting implications. This concept has been reported previously by others. However, rather than using protein concentration changes as a mechanism of regulation, the authors might consider phosphorylation as the mechanism. This represents a much more common method of changing protein-protein interactions. Furthermore, many of the proteins under consideration in this study are known to be phosphorylation substrates. In some cases, the specific sites of phosphorylation are known, and the impact of phosphorylation on protein-protein interactions has been reported. Phosphorylation site mutants are also available. It would be important to understand the relative contributions of phospho-regulation vs. biphasic regulation.

We agree with the reviewer that posttranslational modifications or other interacting partners represent a more common way of modulating protein-protein interactions. We plan to pursue this route through a standalone study. We believe this is clearly beyond the scope of our current manuscript and would require a multi-year effort. Toward the conclusion of our manuscript, we clarified that PTMs may regulate the physiological role of Nde1/Nde1 in the dynein activation pathway:

“We also note that Nde1/Nde1 have been reported to interact with other cellular factors and undergo posttranslational modifications²⁸. While most of the identified phosphorylation sites are located in the C-terminus, phosphorylation of Nde1 at T131 (equivalent to T132 of Nde1) has been shown to reduce its ability to interact with LIS1⁶⁸. The in vitro reconstitution assay we developed could serve as a platform to test how these interaction partners and post-translational modifications regulate the proposed roles of Nde1/Nde1 in the dynein activation pathway.”

Reviewer #3

In this manuscript, Zhao et al sought to assess the assembly and motility of dynein-dynactin-adaptor DDR complex in the presence of Lis1, Nde1, and alpha2 using a TIRF imaging assay. While many of the protein-protein interactions have been previously described – such as between LIS1 and Nde1, between LIS1 and alpha2, between LIS1 and dynein motor domain, between Nde1 and dynein DIC, and between dynactin p150 and dynein DIC – the authors have done a nice job carrying out detailed characterizations, often by titrating one component against the other, to determine how each interaction affects the assembly of DDR complex and ultimately dynein motility. This has provided some new insights into how Nde1 and LIS1 cooperate to trigger the opening of autoinhibited phi dynein in a pathway leading to the assembly of an activated, processive DDR complex. Perhaps the most interesting result is that the authors have found Nde1 to serve as both a positive and negative regulator of dynein motility in a manner dependent on its concentration. They claim that, at low concentrations (e.g., 2 to 20 nM), Nde1 can recruit Lis1 to dynein to promote opening of phi dynein, whereas at high concentrations (e.g., 500 to 2000 nM), Nde1 can compete with dynactin p150 for dynein DIC interaction, thereby negatively regulating the formation of the DDR complex. This claim, if true, appears to offer an explanation to clarify the discrepancy in the literature (e.g., ref. 37, 51, and 30) on whether Nde1 enhances dynein motility or inhibits it.

However, in this reviewer's opinion, the manuscript appears to lack important controls to support their claim, especially with regards to E47A and E47K mutations (DIC-binding mutations in Nde1), causing them to overinterpret their results.

See below our response to Major comment #1.

The mechanism by which excess Nde1 inhibits dynein activation pathway remains elusive. Importantly, although their results provided a possible explanation for why high overexpression of Nde1 is detrimental to dynein function in cells, the physiological significance of the proposed inhibitory effect involving excess Nde1 (i.e., involving excess Nde1 to compete with dynactin p150 for dynein DIC) remains unclear. Whether Nde1 has an inhibitory effect would depend on the physiological concentrations of Nde1 and dynactin p150 in the cell, and their respective affinity (dissociation constant) for dynein IC, which remains unknown.

We agree with the reviewer's view that the higher concentration of Nde1 does not reflect the normal situation in cells. In the original manuscript, we focused on the concept of biphasic regulation from the in vitro mechanistic point of view. These competing effects of Nde1 at different concentration regimes enabled us to provide an explanation for the conflicting results reported in previous in vitro and overexpression studies of Nde1/Nde1, and to potentially end the controversy in the regulatory roles of Nde1. In the revised manuscript, clarified our interpretation that activating role of Nde1 at low concentrations represents its physiological role, whereas its inhibitory role appears at concentrations much higher than the cellular concentration of Nde1, but these observations provide an explanation for some of the conflicting observations previously made in the field.

We clarified these points by 1) changing our title, 2) revising the abstract, and 3) substantially editing the Discussion. For example, in the Discussion of the revised manuscript, we wrote:

“Our model provides a mechanistic explanation for seemingly conflicting observations made for Nde1 and Lis1 in vivo and in vitro. While cell based-assays showed that Nde1 is crucial for the

recruitment of Lis1 to activate dynein 26, 35, 38, 59, 66, several in vitro studies reported inhibition of dynein by Nde1 30, 47. We showed that Nde1 can serve as both the positive and negative regulator of dynein under different concentration regimes. A low concentration of Nde1 is sufficient to tether Lis1 to dynein, increasing its apparent affinity for dynein. We propose that this concentration regime represents the physiological function of Nde1 in the dynein activation pathway. Consistent with this view, cellular studies have shown that Nde1 depletion can be rescued by overexpression of Lis1, because increasing the cellular concentration of Lis1 may be sufficient to stimulate Lis1 binding to dynein and stimulate DDA assembly even in the absence of Nde1 36. However, Nde1 overexpression cannot rescue Lis1 deletion 36, because Nde1 is incapable of activating dynein without Lis1. We also observed that a very high concentration of Nde1 could play an inhibitory role by competing against DIC-dynactin interaction in vitro, providing an explanation for why excess Nde1 has deleterious effects on dynein function in cells 21, 36, 42, 43. We propose that the physiological concentration of Nde1 is lower than this inhibitory regime and dynactin can efficiently displace Nde1 from dynein during the assembly of active complexes.”

Additionally, the conceptual advance of this paper is diluted by the fact that previous studies have already demonstrated a role for Nde1/Ndel1 in tethering Lis1 to dynein (e.g., ref. 37 and 51). Previous studies have also demonstrated that Nde1 competes with p150 subunit of dynactin for interaction with dynein DIC (ref. 31). Thus, in this reviewer’s opinion, the manuscript is not appropriate for publication in its current state.

We thank the reviewer for this comment as it helps us clarify the significant advance our study makes in the field. Studies in live cells suggested that Nde1’s primary function is to tether Lis1 to dynein, increasing its apparent affinity for dynein. However, this prediction could not be directly tested due to the lack of reconstituted dynein activation assay from purified components. In addition, it remained unclear whether Nde1 requires additional factors to tether Lis1 to dynein, or Nde1 and Lis1 can efficiently activate dynein in the absence of other cellular factors. Furthermore, the tethering model was challenged by several observations made in vivo (such as the Lis1 binding mutant of Nde1 to rescue dynein function in Nde1 knockdown conditions). Several in vitro studies also reported that Nde1 inhibits dynein, rather than activating the motor (i.e. Garrot et al. JBC 2023), which seems to be at odds with most of the observations made in vivo. Therefore, the mechanism by which Nde1 promotes dynein function remained highly controversial.

The in vitro reconstitution assay we developed enabled us to directly test the predictions of the tethering model (in the absence of other cellular factors) for the first time. We were also able to observe dynactin binding to stimulate the release of Nde1 from dynein, which we believe is the first step of the dissociation of Nde1 and Lis1 from the complex, after these factors achieve their essential roles in the assembly of the dynein-dynactin complex. We agree with the reviewer that our results are largely consistent with most of the previous studies in live cells, but we see this as its strength rather than its weakness. Our work not only provided direct molecular proof for the tethering model, but also provided an explanation for some of the confusing observations made in vivo and in vitro (such as high overexpression of Nde1 to inhibit dynein function in cells). The assay we developed in this study also provides an opportunity to test the molecular mechanism of other factors in the dynein activation pathway in the future.

Major comments:

1. The author generated E47A and E47K mutations to selectively disrupt DIC-binding of Nde1. However, these Nde1 mutants at 20 nM (i.e., low concentration) failed to boost the wtDDR run frequency in the presence of 100 nM Lis1 (in Fig. 5c). They also failed to decrease the mtDDR run frequency at 500 nM (i.e., high concentration, Fig. 5e). Is it possible that these mutations resulted in a global change in the protein structure, causing unfolding and rendering the protein non-functional? The authors proceeded to conclude that both Lis1-binding and DIC-binding of Nde1 are required for Lis1-mediated activation of dynein, but the data do not fully support this conclusion. Evidence showing that these mutant proteins are folded properly should be provided.

To address this concern, we performed new experiments using point mutants of Nde1. If the mutations do not affect the folding of the protein, we expect the Lis1 binding mutant to bind dynein, and the dynein binding mutant to bind Lis1. Conversely, if the mutations cause a global change in the protein structure, we anticipated these mutants not to bind dynein or Lis1. Mass photometry experiments revealed that the DIC-binding mutants of Nde1 form a complex with Lis1 (indistinguishable from wild-type Nde1). These mutants showed defects in colocalizing with dynein in single molecule assays, strongly indicating that they are properly folded proteins. Similarly, the Lis1 binding mutants maintained their dimerization and their interaction with dynein but did not exhibit Lis1 binding. These results are now shown in the new Supplementary Figure 6 of the revised manuscript.

We wrote: “Mass photometry assays confirmed that the mutations did not disrupt dimerization of Nde1 (Supplementary Fig. 6b). The Lis1 binding mutants (Nde1^{E118A/R129A} and Nde1^{E118K/R129E}) did not form a complex with Lis1, but colocalized with dynein in single molecule colocalization assays (Supplementary Fig. 6c-e). Similarly, the DIC binding mutants (Nde1^{E47A} and Nde1^{E47K}) did not colocalize to dynein (Supplementary Fig. 6c, d), but maintained their association with Lis1 (Supplementary Fig. 6e).”

2. If both Lis1-binding and DIC-binding are required for enhancing wtDDR run frequency in the presence of 100 nM Lis1 (at 20 nM Nde1, see page 7 and Fig. 5c), then what is the role for the interaction of Nde1 with the dynein motor domain in the pathway of forming the DDR complex? The authors seemed to mention that Nde1 can bind dynein through the motor domain (see page 6, second paragraph). A reference for this interaction should be provided. If Nde1 can bind dynein through DIC and/or the motor domain, then it is reasonable to think that motor-domain-binding by Nde1, instead of DIC-binding, could be involved in Step #2 (Nde1 mediating recruitment of Lis1 to dynein) of the model in Fig. 7.

We thank the reviewer for this comment as it helped us clarify the main text.

Previous yeast two-hybrid assays (Sasaki, 2000) and pulldown of cell lysates (Liang, 2004) showed that the C terminus of Nde1 interacts with DHC, but in vitro studies did not find evidence for Nde1 to bind DHC (McKenney et al. 2010). Our results are consistent with McKenney et al. First, we showed that the Nde1 C-terminus is dispensable and the N-terminal coiled coil domain of Nde1 (amino acids 1-190, which lacks the entire C-terminal region) is sufficient to promote Lis1-mediated activation of dynein at low concentration and inhibit DDR assembly at high concentration. Second, we now present experimental evidence that the DIC binding point mutants of Nde1 (E47A and E47K, which contain the C terminal region) do not colocalize to dynein in single molecule colocalization assays and stimulate Lis1-mediated

activation of dynein, demonstrating that the Nde1 C-terminus does not bind dynein in our assay conditions.

In response to this comment, we made several changes to the main text of the revised manuscript:

1. When we first introduced the C-terminal deletion construct of Nde1 (Nde1¹⁻¹⁹⁰), we wrote: “To distinguish whether the N-terminal coiled-coil of Nde1 is sufficient or the C-terminus also contributes to the regulatory role of Nde1 in the dynein activation pathway, we truncated the C-terminus of Nde1 (Nde1¹⁻¹⁹⁰) and determined how it regulates dynein motility.” This section ends with a conclusion that “...the N-terminal coiled-coil is sufficient for both the activating and inhibitory effects of Nde1 on dynein motility.”
 2. We showed that the DIC binding point mutants of Nde1 (E47A and E47K, which contain the C terminal region) do not colocalize to dynein in single molecule colocalization assays. We concluded this section as follows: “Previous studies reported that the C terminus of Nde1 interacts with DHC^{35, 59}, but in vitro studies did not find evidence for Nde1 to bind DHC³⁰. Because we observed a point mutant on the N-terminal coiled-coil of Nde1 to fully disrupt dynein binding, our results confirm that the C terminus of Nde1 does not bind dynein.”
 3. In the section the reviewer is referring to, we removed Nde1 binding to DHC as an alternative possibility of Nde1 mediated inhibition of dynein motility, as this possibility is refuted by the single molecule colocalization assays using the E47A mutant.
3. Run frequency was calculated from the TIRF images and was used as a readout for dynein activation pathway. While this seems like a good way to normalize the number of processive wtDDR on each microtubule for comparison under different conditions, why do the kymographs for the conditions with 2000 nM Nde1 (for example, those in Fig. 1d, 1f, and 2c) consistently show a dramatically smaller number of landing events compared to the conditions with lower concentrations of Nde1? Are these kymographs representative?

Yes, these kymographs are representative of the average behavior. In these motility assays, we fluorescently labeled the cargo adaptor (BicDR1), not dynein. This is because dynein can land onto the microtubule and exhibit diffusive behavior even if it does not assemble a complex with dynactin and BicDR1. Therefore, dynein kymographs do not clearly represent the microtubule landing and motility of DDR complexes. Unlike dynein, BiCDR1 does not bind to microtubules on its own, and only moves on microtubules when it forms a complex with dynein and dynactin. Therefore, BicDR1 imaging better represents DDR motility along microtubules than dynein, as we previously demonstrated (EIShenawy et al. Nat Chem Bio 2019; EIShenawy et al NCB 2020; Canty et al. Nat Comms 2023). The reason why we observe fewer BicDR1 landing events and hence processive runs at high Nde1 concentrations is because there are fewer DDR complexes formed under these conditions.

We also note that previous studies in the field have used either run frequency (the number of processive runs per length of microtubule per second) or landing rate (the number of microtubule landing events per length of microtubule per second) to quantify the motility. We prefer to use run frequency as fluorescent spots that exhibit a unidirectional run along the

microtubule must surely represent active DDR complexes, whereas the ones that land onto the microtubule but do not move at all (or diffuse) may represent incomplete or inactive complexes.

Additionally, the authors should provide more clarity in the Methods section on how they determined the length of the microtubule when calculating the run frequency. Related to adding clarity, Fig. 1a did not specify how or if the surface-immobilized microtubules are fluorescently labeled.

In our earlier studies and at the initiation of the project, we performed single molecule motility assays using fluorescently labeled microtubules and confirmed that we only observe unidirectional processive motility events along microtubules. For the subsequent data collection, we prefer not to use labeled microtubules to eliminate the possibility that labeling of microtubules with organic dyes may interfere with motility. We determined the orientation of unlabeled microtubules by plotting the z-stack images (i.e. maximum values of each pixel in the entire movie into a single frame) of the movies we recorded. The figure on the right shows an example z-stack image in one of our assays. We use these images to calculate the microtubule length and draw segmented lines for kymography analysis.

We wrote in the Methods: “Single molecule motility of DDR was recorded for 500 frames per imaging area and analyzed using the z-stack function of FIJI to determine the orientation of unlabeled microtubules on the surface. Microtubule tracks longer than 10 μm have been included in data analysis.”

We also indicated that we used unlabeled microtubules and labeled BicDR1 (not dynein) in the legend of Fig. 1.

4. The authors claimed that they observed only a small fraction of DDR complexes comigrated with Nde1 in three-color TIRF assays shown in Fig. 4c. However, the kymographs in this figure panel did not make sense. While the authors selectively pointed to tracks (with red arrows) showing comigration of Lis1-LD555 with a processive BicdR1-mNG, they seemed to have ignored tracks showing migration of Lis1-LD555 without BicdR1-mNG that are also visible on the kymographs. Can the authors explain why? The number of migrating Lis1-LD555 tracks without BicdR1-mNG relative to the number of migrating BicdR1-mNG tracks without Lis1-LD555 should be counted and reported. This might help uncover the noise that is inherent in this assay and allow other labs to reproduce their observations.

We perform our three-color imaging assays at relatively low temporal resolution because we use the time sharing mode to prevent crosstalk between different channels. Due to the high velocity of DDR complexes and a relatively low frame rate, processive runs often appear as dashed diagonal lines rather than solid lines in these kymographs. This might give the impression that that there are an unusually high number of Lis1 runs that do not colocalize with

dynein. To address this concern, we plotted the merged image of BicDR1, Lis1, and Nde1 in Figure 4d, which better represents the colocalization between Lis1 and BicDR1 (because gaps between the dashed lines of one color is filled with the other color, appearing as a solid diagonal line with colored stripes in kymographs).

Careful analysis of these kymographs showed that we observe occasional (typically less than 10%) processive runs of Lis1 or Nde1 not colocalizing with dynein due to incomplete labeling of proteins with a dye. The figure on the right shows the percentage of Lis1 and Nde1 runs that do not colocalize with dynein. The velocity of these runs are statistically the same as the runs that colocalize with dynein (not shown). In the revised manuscript, we also highlighted the Lis1 runs that colocalize and not colocalize with BiDR1 with different colored arrows. We also wrote: “Processive runs of Lis1 not colocalizing with BicDR1 (blue arrow) is due to incomplete labeling of proteins with a dye.” in the legend of this panel.

In the Methods, we reported the typical labeling efficiencies of these proteins with their corresponding dyes, as follows: “The probability, p of each SNAP-tagged monomer with benzyl guanine derivatized dyes was 0.65. The probabilities of a SNAP-Lis1 dimer to be labeled with at least one dye (calculated as $2p - p^2$) was 0.88. The labeling efficiency of the ybbR tag was 0.50 and the probability of a dimeric proteins labeled with at least one dye was 0.75.”

5. In Fig. 4a, excess Nde1 was added before or after the assembly of the DDR complex. However, the label on the figure panel says “added during assembly”. To avoid confusion, the authors should clarify the difference between during and before. Alternatively, the authors should use the same wording.

We used the same wording in the revised manuscript.

6. In Fig. 3f, why does increasing full-length Lis1 (from 1 to 2.5 to 10 nM) appear to decrease the association of Nde1(1-190) with dynein in single-molecule colocalization assay? This is inconsistent with Fig. 2f showing that increasing Nde1 would linearly increase the colocalization of Lis1 to surface-immobilized dynein in the same type of assay.

We performed new experiments to address this question. We first tried a 10-fold higher concentration of Nde1¹⁻¹⁹⁰ and still observed an initial increase and then decrease of Nde1 to dynein colocalization as we increased the Lis1 concentration. Next, we performed single molecule colocalization assays using full-length Nde1. As the reviewer predicted, increasing the Lis1 concentration promotes full-length Nde1 localization to dynein when we used either 1 nM or 10 nM Nde1. These new results are now shown in the new Figure 2c-d.

We are still not sure why we observe different behavior when we used Nde1¹⁻¹⁹⁰. We noticed that this construct has higher affinity to Lis1 than full-length Nde1 and it is more likely to recruit two Lis1s than full length Nde1 (Fig. 2f). We also observed Lis1 and Nde1 to stabilize each other's binding to dynein (new Fig. 2e). We speculate that, when Nde1¹⁻¹⁹⁰ simultaneously binds

to two Lis1 dimers, it can still bind to DIC but cannot form a ternary complex with Lis1. This may lead to a quicker release of Nde1 from dynein, reducing the percentage of dynein motors that colocalize with Nde1. Because these ideas are highly speculative at this stage, we did not discuss them in the revised manuscript, and moved the results of the single molecule colocalization assays of Nde1¹⁻¹⁹⁰ to Supplementary Figure 2b. We also noted that this behavior is different from full-length Nde1 in the legend of that figure.

Minor comments:

7. Typo on page 3, last paragraph of Introduction, 5th line: “Nde11” should be “Nde1”.

Fixed.

8. Error on page 9, middle paragraph, last sentence: “overexpression of Nde1....” is repeated twice.

Fixed.

9. Error in a sentence on page 7: “Lis1 has been initially identified as the noncatalytic β subunit of PAF-AH and the catalytically-active $\alpha 1$ and $\alpha 2$ subunits of PAF-AH to inactivate dynein by interacting with Lis1”. This sentence implies that Lis1 has been initially identified as alpha2 subunit of PAF-AH.

Fixed.

REVIEWER COMMENTS

Reviewer #2 (Remarks to the Author):

The authors have responded appropriately to the concept of biphasic regulation of dynein by Nde1. Although some of the original language about alternating roles for Nde1 remains, they do a better job of focusing on the lower concentration of Nde1 as the more important role.

There remain two important transitions that the authors do not address sufficiently.

1) The ability of Nde1 to "rescue" LIS-1 from PAFAH complexes is emphasized but not addressed. How does Nde1 "rescue" LIS-1 from PAFAH alpha to allow binding to dynein? Understanding how, when and why this happens would be an important aspect of this mode that needs data to support.

2) The switch of dynein IC binding from Nde1 to dynactin is also not addressed. Given the importance of this step in the model, I think some experiments are needed to provide insight. The Nde1 concentration aspect of this suggests simple competition. However, because this does not happen in cells, another mechanism must be used. In the absence of data, this is not explained sufficiently.

Reviewer #3 (Remarks to the Author):

The authors have addressed all my comments and questions satisfactorily. In particular, the new experiments that they have performed (in the new Supplementary Figure 6) have provided evidence indicating that the DIC binding mutants (E47A and E47K) are properly folded proteins.

We want to thank the reviewers for carefully reviewing our revised manuscript. We are glad to see that the reviewers are mostly satisfied with the revision. Reviewer 2 has raised valid remaining concerns, which helped us further support our conclusions with new experimental evidence and clarify our model. A detailed explanation of these revisions can be found in our point-by-point response (highlighted in blue) to the comments (highlighted in black) below. We also highlighted the major changes to the manuscript file in yellow.

REVIEWER COMMENTS

Reviewer #2:

The authors have responded appropriately to the concept of biphasic regulation of dynein by Nde1. Although some of the original language about alternating roles for Nde1 remains, they do a better job of focusing on the lower concentration of Nde1 as the more important role.

There remain two important transitions that the authors do not address sufficiently.

1) The ability of Nde1 to "rescue" LIS-1 from PAFAH complexes is emphasized but not addressed. How does Nde1 "rescue" LIS-1 from PAFAH alpha to allow binding to dynein? Understanding how, when, and why this happens would be an important aspect of this mode that needs data to support.

To address this concern, we provided new experimental evidence and substantially revised the sections that cover Nde1 and $\alpha 2$ in Results and Discussion.

In Results, we wrote:

"In addition to its regulatory role in the dynein activation pathway, Lis1 also serves as the noncatalytic β subunit of the PAF-AH1B complex in vertebrates⁵⁸ and can modulate PAF-AH1B enzyme activity in vitro⁵⁹. It remains mysterious whether Lis1's roles in these two regulatory pathways are coupled together, but the removal of the α subunit of PAF-AH1B did not result in defects in neurodevelopment⁶⁰, suggesting that the Lissencephaly phenotype is distinct from Lis1's role in the PAF-AH1B pathway. Overexpression of catalytic $\alpha 1$ or $\alpha 2$ subunits of PAF-AH1B has been shown to result in the inactivation of dynein-driven processes, whereas further overexpression of Lis1 or Nde1 restored dynein function in cells⁶¹. Therefore, Nde1 and $\alpha 1/\alpha 2$ appear to compete for recruiting available Lis1 in the cytosol."

In the previous version, we used mass photometry to study the complex formation of Nde1-Lis1 and $\alpha 2$ -Lis1 and showed that $\alpha 2$ and Nde1 cannot simultaneously bind to the same Lis1 dimer. However, because the Nde1 and $\alpha 2$ constructs we used had similar masses, we could not directly distinguish which of these proteins formed a complex with Lis1 from mass measurements. In this second revision, we performed size exclusion experiments using Lis1, Nde1, and $\alpha 2$ labeled with three different colors and imaged elutions from the column using a Typhoon fluorescent gel scanner (similar to Tarricone et al 2004). We wrote:

"Because Nde1¹⁻¹⁹⁰ and $\alpha 2$ constructs we used had similar masses, which of these proteins formed a complex with Lis1 could not be distinguished from mass measurements (Fig. 6b). We used size exclusion experiments, which enabled us to resolve the peaks of Nde1-Lis1 and $\alpha 2$ -

Lis1 due to the elongated shape Nde1¹⁻¹⁹⁰ (Fig. 6c and Supplementary Fig. 7). Nde1 and $\alpha 2$ formed a complex with Lis1 at equal ratios when mixed at equal concentrations. Increasing the relative concentration of Nde1 favored the formation of Nde1-Lis1, and similarly, increasing the concentration of $\alpha 2$ favors the formation of $\alpha 2$ -Lis1 (Figure 6c), suggesting that Nde1 and $\alpha 2$ have similar affinities to bind Lis1¹⁴.”

We also performed new single-molecule colocalization assays, which showed that increasing the concentration of Nde1 reduces the colocalization of $\alpha 2$ to surface-bound Lis1. In the previous version of our manuscript, we also showed that $\alpha 2$ blocks Lis1-mediated activation of dynein in the absence of Nde1, and the addition of Nde1 in an equal amount to $\alpha 2$ does not only enable Lis1 binding to dynein, but it also further boosts Lis1’s ability to activate dynein.

We thank the reviewer for this comment as it helped us clarify our model. We replaced “Nde1 rescues Lis1 from $\alpha 2$ inhibition” with “Nde1 competes for the same cytosolic pool of Lis1 with the subunits of the PAF-AH1B complex”. We also added, “Without Nde1/Ndel1, Lis1 will be bound to PAF-AH1B subunits and unable to interact with dynein.” We also clarified our thinking about an important question in the field: Why would a cell need another factor (Nde1) if Lis1 can bind and activate dynein on its own? We propose that 1) Nde1 is needed to recruit Lis1 to autoinhibited dynein, in which the Lis1 binding site is buried but the Nde1 binding site is exposed. 2) Nde1 is also needed to compete against $\alpha 1/\alpha 2$ subunits of PAF-AH for Lis1. Otherwise, Lis1 will be bound to PAF-AH subunits and unable to interact with dynein.

In Discussion, we wrote:

“The assembly of processive DDA complexes requires the opening of phi dynein⁶, which is primarily driven by Lis1 binding to the AAA+ ring^{11, 16-18, 40}. Based on our results and previous reports, we provide a mechanistic explanation for why another cellular factor (Nde1/Ndel1) is needed if Lis1 can bind and activate dynein on its own. We propose that Nde1/Ndel1 has two major roles in the dynein activation pathway (Fig. 7). First, Nde1/Ndel1 competes against $\alpha 1/\alpha 2$ subunits of PAF-AH1B for Lis1. Because haploinsufficiency of Lis1 is sufficient to disrupt dynein function in cells³⁸ and cause disease¹¹, Lis1 is likely to be present in limited amounts in the cell and Nde1/Ndel1 and $\alpha 1/\alpha 2$ may compete for cytosolic Lis1 for their regulatory roles in dynein and PAF-AH1B pathways, respectively. In this case, the absence of Nde1/Ndel1 would result in the sequestration of Lis1 to PAF-AH1B and, therefore, blocking Lis1’s access to dynein⁶¹...”

As the reviewer pointed out, how and when Nde1-PAF-AH competition happens, and why Nde1 can hand over Lis1 to dynein, whereas $\alpha 1/\alpha 2$ does not, still remain unclear and will require future detailed investigations both in vivo and in vitro.

2) The switch of dynein IC binding from Nde1 to dynactin is also not addressed. Given the importance of this step in the model, I think some experiments are needed to provide insight. The Nde1 concentration aspect of this suggests simple competition. However, because this does not happen in cells, another mechanism must be used. In the absence of data, this is not explained sufficiently.

This is another excellent question raised by the reviewer. In our re-revised manuscript, we provide new experimental data to provide a possible explanation for why Nde1 and dynactin do not inhibit each other despite having overlapping binding sites on DIC. We wrote:

“If Nde1 and dynactin binding to DIC are mutually exclusive, we anticipated dynactin binding to release Nde1 from dynein. Because dynactin has a low affinity to bind phi dynein and more readily interacts with open dynein⁶, we tested this possibility for both mtDyn, which cannot form the phi conformation and remains in the open conformation⁶ and wtDyn, which contains a mixture of phi (~70%) and open conformation (~30%, not shown). We first decorated surface-immobilized microtubules with mtDyn and observed Nde1 binding to mtDyn on microtubules (Fig. 4e). The addition of dynactin, but not the cargo adaptor, caused almost all of Nde1 to be released from microtubules (Fig. 4e, f). Second, we monitored Nde1 colocalization to surface-bound dynein. The addition of dynactin reduces Nde1 colocalization to wtDyn by 37% (Fig. 4g). When we repeated the same experiment with mtDyn, dynactin addition substantially reduced the colocalization of Nde1 from dynein by 75% (Fig. 4g). These results are consistent with the idea that dynactin causes the release of Nde1 from DIC after it interacts with dynein in the open conformation.”

We propose that Nde1 binds to DIC and recruits Lis1 to phi dynein. Lis1 opens the phi conformation. Dynactin is known to interact with open dynein but has a low affinity to bind phi dynein. Therefore, dynactin is more likely to cause the release of Nde1 after the opening of phi conformation (i.e. after Nde1 has accomplished its role in the dynein activation pathway).

We wrote:

“It remains to be determined why Nde1 and dynactin do not inhibit each other in cells despite their overlapping binding sites on DIC. One possible explanation is that Nde1 and dynactin perform their regulatory roles in two distinct conformations of dynein. According to this scheme, dynactin does not prevent recruitment of Nde1 to phi dynein because it does not strongly interact with dynein in this conformation⁶. Dynactin binds dynein and causes dissociation of Nde1 from DIC after the opening of the phi conformation by Nde1/Lis1. Therefore, the physiological concentration of Nde1 does not prevent the association of open dynein with dynactin. Predictions of this model and the order of events that leads to the activation of the dynein transport machinery need to be tested by future structural and biophysical studies.”

REVIEWERS' COMMENTS

Reviewer #2 (Remarks to the Author):

In this revision, the authors make some changes that improve the study overall. However, several issues remain that have the potential to confuse the audience. It would be beneficial to the reader to clarify these points.

1) This argument that Nde1 has two alternative functions remains in the manuscript. As I think the authors have shown, Nde1 has only one function at physiological concentrations. The inhibitory effect of unnaturally high concentrations of Nde1 is an artifact of in vitro assays and is not a true function for Nde1. Although this effect was observed in vitro and has been reported, it should be referred to as an artifact rather than a natural function for Nde1. The confusing nature of this distinction is apparent in the Abstract where three functions for Nde1 are listed: 1) competition with PAFAH, 2) recruitment of Lis1 to dynein, and 3) inhibition of dynein.

2) The role of Nde1 in separating a fraction of Lis1 from PAFAH is a distraction and potentially weakens the explanation of Nde1's role in recruiting Lis1 to dynein. The ratiometric measurements suggest that roughly half of Lis1 is part of PAFAH and the other half is involved in dynein activation. Does this mean that one half of the Lis1 in cells is sufficient to promote dynein motility? Or are there conditions under which Lis1's role in dynein regulation is preferred over its role in PAFAH activity (and visa versa)? In the absence of a better understanding of this dynamic, it is not clear that this strengthens the manuscript (and potentially weakens it).

3) The sequence of events for the competition in binding between Nde1 and dynactin for the DICs is not presented consistently. Does dynactin displace Nde1 through some unknown mechanism, or does it simply have higher affinity? Or is Nde1 released allowing dynactin to gain access to the DICs? The authors describe these transitions in different ways in the manuscript, which is confusing. Do the open and closed configurations of the dynein heads also affect the structure of the DICs, thereby biasing the binding to Nde1 or dynactin?

Minor points:

1) I did not find any data relevant to the point about dynactin recruiting multiple dynein complexes in the manuscript (page 4). Are the authors referring to previous work by others?

2) The Abstract needs improvement for flow.

RESPONSE TO REVIEWERS

We thank Reviewer 2 for providing additional comments to improve the main message of our paper. Below, we provide point-by-point responses to the criticisms and suggestions of the reviewer.

Reviewer #2 (Remarks to the Author):

In this revision, the authors make some changes that improve the study overall. However, several issues remain that have the potential to confuse the audience. It would be beneficial to the reader to clarify these points.

1) This argument that Nde1 has two alternative functions remains in the manuscript. As I think the authors have shown, Nde1 has only one function at physiological concentrations. The inhibitory effect of unnaturally high concentrations of Nde1 is an artifact of in vitro assays and is not a true function for Nde1. Although this effect was observed in vitro and has been reported, it should be referred to as an artifact rather than a natural function for Nde1. The confusing nature of this distinction is apparent in the Abstract where three functions for Nde1 are listed: 1) competition with PFAH, 2) recruitment of Lis1 to dynein, and 3) inhibition of dynein.

We agree with the reviewer that the inhibitory effect of Nde1 only occurs at very high concentrations. We made this point clear in the Discussion of our previous revision. We also made it clear that these observations provide a possible explanation for some of the earlier observations made in cells under overexpression conditions and for differences between our observations and Garrot et al. JBC 2023, which only observed the inhibitory role of Nde1 on dynein motility in vitro.

In this final version, we do not mention this observation in the Abstract to avoid confusion.

2) The role of Nde1 in separating a fraction of Lis1 from PFAH is a distraction and potentially weakens the explanation of Nde1's role in recruiting Lis1 to dynein. The ratio measurements suggest that roughly half of Lis1 is part of PFAH and the other half is involved in dynein activation. Does this mean that one half of the Lis1 in cells is sufficient to promote dynein motility? Or are there conditions under which Lis1's role in dynein regulation is preferred over its role in PFAH activity (and visa versa)? In the absence of a better understanding of this dynamic, it is not clear that this strengthens the manuscript (and potentially weakens it).

We disagree with this view. The data we presented in this manuscript and earlier observations provide a possible explanation for why a cell needs another factor(s) if Lis1 can activate dynein on its own.

- i. Lis1 cannot stably bind to autoinhibited dynein and Nde1/Nde1 tethers Lis1 to dynein, locally enriching its concentration near the dynein motor domain for Lis1 to efficiently open the phi conformation. As the reviewer states, this is the primary function of Nde1/Nde1.
- ii. Lis1 is in limited quantities in cells, as haploinsufficiency of Lis1 causes disease. In the absence of Nde1/Nde1, it is possible that cytosolic Lis1 is recruited to PFAH as a noncatalytic subunit and is not able to interact with dynein.

Therefore, Nde1/Ndel1 does not only increase the apparent affinity of Lis1 for dynein, but it also secures one-half of cytosolic Lis1 for the dynein pathway.

Because only one-half of cellular Lis1 interacts with dynein, one-half of the cellular concentration of Lis1 is likely sufficient for dynein activation. We still do not know whether there is an overlap between these two functions of Lis1 or whether cells favor one role over the other by posttranslational modifications or by additional cofactors. These possibilities will need to be tested in future studies.

Following editorial suggestions, we tone down our statements about the potential cellular role of Nde1 and alpha2 competition to bind Lis1.

3) The sequence of events for the competition in binding between Nde1 and dynactin for the DICs is not presented consistently. Does dynactin displace Nde1 through some unknown mechanism, or does it simply have higher affinity? Or is Nde1 released allowing dynactin to gain access to the DICs? The authors describe these transitions in different ways in the manuscript, which is confusing. Do the open and closed configurations of the dynein heads also affect the structure of the DICs, thereby biasing the binding to Nde1 or dynactin?

The mechanism by which dynactin displaces Nde1 from DIC requires its own investigation. Because this binding site is located in a flexible extension of DIC and not seen in available structural studies, we believe that it is accessible in both open and closed conformations of dynein and that Nde1 can bind to the DIC N-terminus in both conformations. Dynactin interacts with the DIC N-terminus via the coiled-coil extension of its p150 subunit. A recent cryo-EM study (Lau et al. *Embo J* 2021) showed that the p150 arm of isolated dynactin folds back and docks onto the pointed end of dynactin. It is possible that p150 cannot interact with Nde1 in this docked conformation and the binding of open dynein and a cargo adaptor to dynactin result in the undocking of the p150 arm and facilitate its interaction with DIC. p150 may displace Nde1 either because it has a higher affinity for DIC or it is stably tethered to dynein via dynactin whereas Nde1 remains untethered after it hands off Lis1 to dynein. As a result, Nde1 can tether Lis1 to closed dynein without competing against dynactin and dynactin can displace Nde1 after it stably binds to the DHC of open dynein.

We revised the Discussion to further clarify these possibilities and avoid confusion.

“It remains to be determined why Nde1 and dynactin do not inhibit each other in cells despite their overlapping binding sites on DIC. One possible explanation is that Nde1 and dynactin perform their regulatory roles in two distinct conformations of dynein. According to this scheme, dynactin does not prevent recruitment of Nde1 to phi dynein because it does not strongly interact with dynein in this conformation⁶. The p150^{Glued} arm of isolated dynactin is observed to be folded back and docked onto the pointed end of dynactin, blocking the binding site of cargo adaptors⁶⁸. Docking of the p150^{Glued} arm may also prevent its interaction with DIC. Recruitment of open dynein and a cargo adaptor to dynactin results in undocking of the p150^{Glued} arm⁶⁸, which may activate p150^{Glued}'s interaction with DIC. As a result, dynactin may trigger the dissociation of Nde1 from DIC after the opening of phi dynein.”

Minor points:

1) I did not find any data relevant to the point about dynactin recruiting multiple dynein complexes in the manuscript (page 4). Are the authors referring to previous work by others?

We refer to the previous work we published in this paragraph. In earlier studies (Urnavicius et al. Nature 2017 and ElShenawy et al. Nat Chem Bio 2019), we have shown that dynactin can recruit two dyneins, and this results in faster motility of the complex. More recent studies in our laboratories (references 16 and 17) showed that Lis1 facilitates the recruitment of two dyneins to dynactin, thereby increasing the speed of the complex. In this study, we observed that the addition of Nde1 slightly increases the average speed of dynein-dynactin-adaptor complexes, strongly suggesting that Nde1 further promotes the formation of complexes with two dyneins.

2) The Abstract needs improvement for flow.

We revised the abstract as follows:

“Cytoplasmic dynein drives the motility and force generation functions towards the microtubule minus end. The activation of dynein motility requires its assembly with dynactin and a cargo adaptor. This process is facilitated by two dynein-associated factors, Lis1 and Nde1/Ndel1. Recent studies proposed that Lis1 relieves dynein from its autoinhibited conformation, but the physiological function of Nde1/Ndel1 remains elusive. Here, we investigated how human Nde1 and Lis1 regulate the assembly and subsequent motility of mammalian dynein using in vitro reconstitution and single molecule imaging. We found that Nde1 recruits Lis1 to autoinhibited dynein and promotes Lis1-mediated assembly of dynein-dynactin-adaptor complexes. Nde1 also competes against the recruitment of Lis1 to platelet activator factor acetylhydrolase (PAF-AH) 1B as a noncatalytic subunit, thus promoting Lis1 binding to dynein. Before the initiation of motility, the association of dynactin with dynein triggers the dissociation of Nde1 from dynein, presumably by competing against Nde1 binding to the dynein intermediate chain. Our results provide a mechanistic explanation for how Nde1 and Lis1 synergistically activate the dynein transport machinery.”